# The mitochondrial methylation potential gates mitoribosome assembly

Ruth I. C. Glasgow[1,8], Vivek Singh [1,2,8], Lucía Peña-Pérez [3,4], Alissa Wilhalm[1], Marco F. Moedas[1,3], David Moore[1], Florian A. Rosenberger [1,4], Xinping Li[5], Ilian Atanassov [5], Mira Saba[1], Miriam Cipullo[1], Joanna Rorbach [1], Anna Wedell[3,4], Christoph Freyer [1,3,9] ✉, Alexey Amunts [6,7,9] ✉ & Anna Wredenberg [1,3,9] ✉

S-adenosylmethionine (SAM) is the principal methyl donor in cells and is essential for mitochondrial gene expression, influencing RNA modifications, translation, and ribosome biogenesis. Using direct long-read RNA sequencing in mouse tissues and embryonic fibroblasts, we show that processing of the mitochondrial ribosomal gene cluster fails in the absence of mitochondrial SAM, leading to an accumulation of unprocessed precursors. Proteomic analysis of ribosome fractions revealed these precursors associated with processing and assembly factors, indicating stalled biogenesis. Structural analysis by cryo-electron microscopy demonstrated that SAM-dependent methylation is required for peptidyl transferase centre formation during mitoribosome assembly. Our findings identify a critical role for SAM in coordinating mitoribosomal RNA processing and large subunit maturation, linking cellular methylation potential to mitochondrial translation capacity.

S-adenosylmethionine (SAM) is a product of one-carbon metabolism and the primary biological methyl group donor within our cells[1]. Its levels are implicated in numerous cellular processes, including the cell's epigenetic state and as a substrate for the biosynthesis of various macromolecules. SAM is synthesised in the cytosol from adenosine 5′-triphosphate (ATP) and methionine as part of the methionine cycle and is subsequently transported to different subcellular compartments. Approximately 30% of cellular SAM resides inside mitochondria, acting as a cofactor for ubiquinone and lipoic acid biosynthesis and as a methyl group donor for mitochondrial RNAs and various proteins[2]. Mitochondrial gene expression especially relies on mitochondrial SAM (mitoSAM), with multiple nucleotide modifications, such as base or 2′-O-ribose methylations, reported on mitochondrial transfer and ribosomal RNAs[3,4]. Additionally, we previously characterised the mitochondrial methylproteome, identifying over a dozen of methylated mitochondrial proteins involved in translation[5].

The mammalian mitochondrial genome (mtDNA) is expressed as long, polycistronic transcripts that cover almost the entire genome length. The canonical mode of mitochondrial RNA processing describes how tRNAs act as recognition sites for dedicated nucleases to sequentially release the flanking transcripts for further maturation in a 5′ to 3′ order[6]. The molecular and structural details of this process have been described, with the mitochondrial ribonucleases (mtRNase) P and Z cleaving the 5′ and 3′ tRNA junctions, respectively[7–11]. mtRNase P consists of the two tRNA-binding subunits, MRPP1 and 2, and the catalytic subunit MRPP3, also known as PRORP[11]. MRPP1, also known as RNA methyltransferase 10 homologue C, TRMT10C, additionally catalyses the formation of N(1)-methylguanine (m1G) and N(1)-

[1]Department of Medical Biochemistry and Biophysics, Karolinska Institutet, Stockholm, Sweden. [2]Science for Life Laboratory, Department of Biochemistry and Biophysics, Stockholm University, Solna, Sweden. [3]Centre for Inherited Metabolic Diseases, Karolinska University Hospital, Stockholm, Sweden. [4]Department of Molecular Medicine and Surgery, Karolinska Institutet, Stockholm, Sweden. [5]Max-Planck Institute for Biology of Ageing, Cologne, Germany. [6]University of Münster, Münster, Germany. [7]Department of Structural Biochemistry, Max Planck Institute of Molecular Physiology, Dortmund, Germany. [8]These authors contributed equally: Ruth I. C. Glasgow, Vivek Singh. [9]These authors jointly supervised this work: Christoph Freyer, Alexey Amunts, Anna Wredenberg. ✉e-mail: christoph.freyer@ki.se; alexey.amunts@gmail.com; anna.wredenberg@ki.se

methyladenine (m[1]A) at position 9 in 19 out of 22 mitochondrial tRNAs[3,12]. Recent structural work, though, indicates that this methylation is not required for cleavage by either mtRNase P or Z[10,13,14].

At least eight residues of the mammalian small (12S) and large (16S) mitochondrial ribosomal subunit RNAs are methylated[15–17]. The responsible SAM-dependent methyltransferases have been identified, with TRMT2B, METTL15, NSUN4 and TFB1M modifying residues of 12S[18–24] and MRM1, MRM2 and MRM3 responsible for 2′-O-ribose methylations on 16S[25–27]. In mice, 12S rRNA only contains base methylations (m[5]U499, m[4]C909, m[5]C911, m[6]₂A1006, m[6]₂A1007), while 16S rRNA only has 2′-O-ribose methylations (Gm2253, Um2481, Gm2482) (mouse mtDNA (NC_005089) numbering shown). Human 16S rRNA contains an additional methylation at m[1]A946 by TRMT61B[28].

The roles of these modifications are not always clear, but the recent structure of a fully modified human mitoribosome shows that six mitochondria-specific modifications help stabilise the decoding centre and facilitate the binding of the mRNA and tRNA ligands[24]. For instance, cytosine N[4]- and N[5]-methylation (m[4]C and m[5]C) by METTL15 and NSUN4 on 12S rRNA have recently been implicated in monosome assembly and translation[21,23,24,29,30]. These modifications interact with the phosphate group of m[4]C1486 (m[4]C909 in mice), thereby stabilising an mRNA kink between the A- and P-sites[21,23,24].

Additionally, on 16S rRNA, 2′-O-ribose methylations by MRM1-3 position the rRNA bases sterically to enhance interactions with the A-tRNA nucleotides[24]. Deleting the bacterial MRM2 homologue, rlmE, delayed ribosome assembly and reduced translation, indicating structural and mechanistic importance for MRM2[31–33]. In contrast, MRM2 methylation activity has also been considered dispensable for ribosome assembly in HEK 293 T cells[34].

SAM cannot be de novo synthesised inside mitochondria and must be imported via the mitochondrial SAM carrier (SAMC), encoded by SLC25A26/Samc[2,35–37]. Variants in SLC25A26 can cause severe mitochondrial disease in humans[38–40], while Samc deletion in mice and flies results in embryonic and larval lethality[5]. We previously replicated patient variants in fly models, demonstrating that a reduced SAM import into mitochondria can result in a progressive mitochondrial gene expression defect, but the mechanism remains unknown[5].

To investigate the undefined importance of mitochondrial RNA methylation we attenuated the mitochondrial methylation potential in a conditional skeletal muscle-specific mouse model and mouse embryonic fibroblasts (MEFs). We show that efficient processing of the mitoribosomal gene cluster depends on mitoSAM. We also demonstrate that the assembly of the small mitoribosomal subunit (mtSSU) begins before the complete processing of 12S rRNA. Finally, our data reveal methylation as a critical peptidyl transferase centre (PTC) assembly and the large mitoribosomal subunit (mtLSU) maturation checkpoint, highlighting a fundamental role for SAM during mitochondrial gene expression.

## Results

### Mitochondrial SAM depletion causes a progressive defect in gene expression

We previously reported that SAMC is the only route for SAM into mitochondria, and its deletion drastically depletes mitochondria of SAM[5]. While the in vivo deletion is lethal and presents a progressive mitoSAM deficiency, Samc KO MEFs (KO[MEF]) can be maintained in culture and represent a chronic state of mitoSAM depletion. To study the progressive decline in mitochondrial methylation potential, we deleted Slc25a26 specifically in mouse skeletal muscle, using Cre recombinase under the control of the myosin light chain 1 f promoter (Mlc1f-Cre). Muscle-specific Samc knockout (KO[SkM]) mice were born at Mendelian ratios and appeared normal at weaning but failed to gain weight compared to their control littermates, establishing an experimental endpoint at 12 weeks of age. We measured m[4]C909 and m[5]C911 methylation of 12S rRNA by bisulfite sequencing as a readout of the

mitochondrial methylation potential. At 8 weeks of age, quadricep muscle exhibited a 12S rRNA methylation level of less than 50% in comparison to controls, with a further decrease by 12 weeks (Fig. 1a). KO[MEF] cells only retained residual methylation at both sites, confirming the lack of methylation potential in these cells.

Quantitative proteomics, comparing control to KO[SkM] quadriceps at 8 and 12 weeks of age, revealed a mild but progressive decline in individual oxidative phosphorylation (OXPHOS) subunit proteins of complexes I, III and IV and a slight increase in complex II and V subunits with age (Fig. 1b and Supplementary Data 1). The increased steady-state levels of complex II and V subunits suggest enhanced mitochondrial biogenesis, consistent with findings from other mouse models and patients with mitochondrial dysfunction. In contrast, KO[MEFs], representing a chronic state of mitoSAM depletion, showed a generalised decrease in OXPHOS subunits, with complexes I and IV most affected[5]. Assessment of mt-mRNA transcript levels ruled out a depleted transcript pool as the cause for the decreased OXPHOS composition, as steady-state levels of most transcripts were increased in KO[SkM] quadriceps at 8 weeks of age and further rose by 12 weeks (Fig. 1c). KO[MEFs] also exhibited elevated steady-state levels, with mtNd1 most changed. Only 12S, 16S and mtCytb RNA levels decreased to ~50% of controls[5].

We next investigated mt-tRNA steady-state levels in quadriceps and MEFs, as most mt-tRNAs are post-transcriptionally methylated, including the m[1]A methylation at position 9 by TRMT10C[3,11,12]. Northern blot analysis revealed a very similar pattern in both models, although the responses to the loss of methylation of individual mt-tRNAs varied drastically (Fig. 1d and Supplementary Fig. 1). For instance, while the steady-state levels for mtR, mtE, mtW and mtV were almost undetectable, mtN, mtC, mtH, mtL1, mtL2, mtS1, mtK and mtY were increased, irrespective of their strand or genome position.

### The mitochondrial methylation potential affects the processing of primary transcripts

We tested whether mitoSAM is required during canonical transcript processing by performing direct long-read RNA sequencing by Oxford nanopore technology (ONT sequencing) on polyadenylated total RNA from 8- and 12-week-old mouse quadriceps and mitochondrial RNA extracts from MEF samples (Supplementary Data 2) (see 'materials and methods'). Reads were aligned to the murine mitochondrial genome (NC_005089), resulting in comparable counts within the muscle and MEF groups (Supplementary Fig. 2 and Supplementary Data 3–9). Fully processed transcripts were defined as starting and ending within ±20 nucleotides of their annotated boundaries, including known untranslated regions (UTRs) and the bicistrons of mtAtp8/6 and mtNd4/4 L (Supplementary Fig. 3a)[41]. Based on this definition, we observed no significant evidence of transcripts initiating or terminating at previously unidentified sites (Supplementary Fig. 3b).

In control samples, most reads aligned with these defined boundaries, consistent with correctly processed mature transcripts (Fig. 2a). As anticipated from the mammalian mode of polycistronic mitochondrial transcription[42], we also detected several processing intermediates, including bi- and polycistronic reads, with some spanning nearly the entire heavy strand. Overall, though, the number of unprocessed transcripts was relatively low across all control samples, accounting for only 1.8% to 3.6% of total reads. This fraction increased in 8-week-old KO[SkM] quadriceps (6.4% versus 1.8% in controls) and further in 12-week-old KO[SkM] quadriceps (7.9% versus 2.3%). In KO[MEFs], unprocessed transcripts accounted for more than one-third of all reads (39.2% versus 3.6% in controls), indicative of a severe processing defect in mitoSAM-depleted samples (Fig. 2a).

We next assessed whether the observed increase in unprocessed transcripts was uniform across all polycistronic species, by evaluating the composition of the unprocessed read pool (Fig. 2b). In control quadriceps, approximately half of all unprocessed reads corresponded

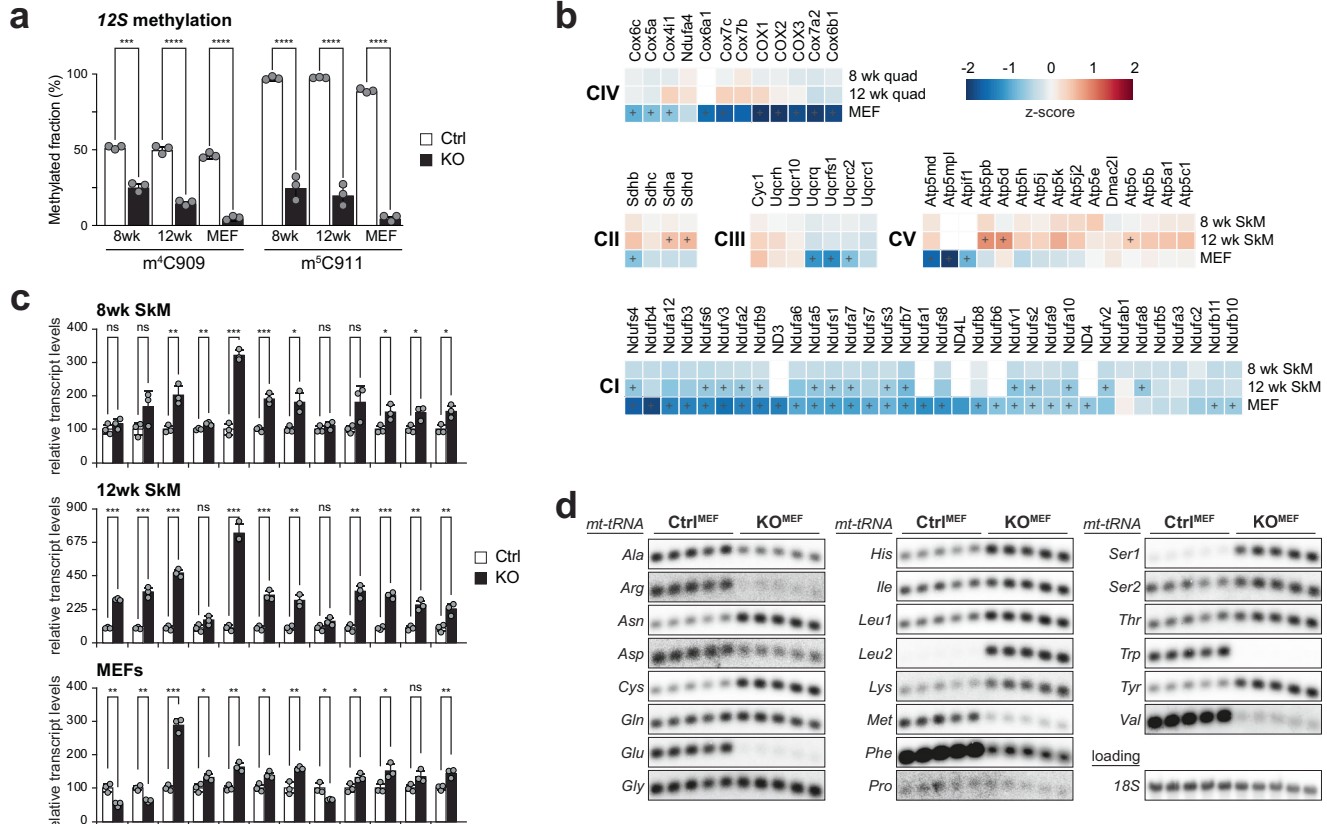

**Fig. 1 | Reduced mitochondrial methylation potential affects mitochondrial gene expression. a** The mitochondrial methylation potential was assessed via bisulfite pyrosequencing of *12S* mt-rRNA from control (white) and *Samc* KO (black) MEFs or quadriceps at 8 and 12 weeks of age, targeting 4'-methylcytosine m⁴C909 (m⁴C) or 5'-methylcytosine m⁵C911 (m⁵C). Data are presented as mean values ± standard deviation. Student's two-tailed T-test was used. ***$p < 0.001$, ****$p < 0.0001$ ($n = 3$). Exact *p* values: m⁴C909: 8 week $p = 0,000139$, 12 week $p = 0.000039$, MEF $p = 0,000005$. m⁵C911: 8wk $p = 0.000096$, 12wk $p = 0.000045$, MEF $p = <0.000000$. **b** Heatmap of OXPHOS subunits as determined by mass spectrometry-based proteomics in 8 and 12-week-old quadriceps or MEF samples normalised to controls ($n = 3$). **c** Relative mitochondrial transcript steady-state levels in 8 and 12-week-old muscle or MEFs from control (white) and *Samc* KO (black) samples, as determined by qRT-PCR. Data are presented as mean values ± standard deviation. Student's two-tailed *T*-test was used. ns = non-significant $p > 0.05$; *$p < 0.05$, **$p < 0.01$, ***$p < 0.001$.($n = 3$ biologically independent samples with 3 technical replicates). Exact p-values: 8wk SkM: 12S $p = 0.276355$, 16S $p = 0.100339$, Nd1 $p = 0.005228$, Nd2 $p = 0.009903$, Nd3 p = 0.000106, Nd4 $p = 0.000978$, Nd5 $p = 0.013336$, Cytb $p = 0.305520$, Atp8/6 $p = 0.060454$, Co1 $p = 0.031159$, Co2 $p = 0.023099$, Co3 $p = 0.018100$. 12 week SkM: 12S $p = 0.000002$, 16S $p = 0,000315$, Nd1 $p = 0.000032$, Nd2 $p = 0,086241$, Nd3 $p = 0.000114$, Nd4 $p = 0,000462$, Nd5 $p = 0,002247$, Cytb $p = 0.088495$, Atp8/6 $p = 0.001081$, Co1 p = 0.000087, Co2 $p = 0,002090$, Co3 $p = 0.008890$. MEF: 12S $p = 0.005025$, 16S $p = 0.001332$, Nd1 $p = 0.000149$, Nd2 $p = 0.030912$, Nd3 $p = 0.002983$, Nd4 $p = 0.012898$, Nd5 $p = 0.006421$, Cytb $p = 0.018639$, Atp8/6 $p = 0.033871$, Co1 $p = 0.030329$, Co2 $p = 0.050202$, Co3 $p = 0.006753$. **d** Mitochondrial tRNA steady-state levels in *Samc* KO or control MEFs, as determined by Northern blot analysis ($n = 5$ biologically independent samples). Source data are provided as a Source Data file.

to the previously described tricistron of *mtAtp8/6-mtCox3*, which is processed independently of the canonical machinery, likely involving members of the FASTK family[43–48]. This transcript was also detectable in control MEFs, albeit at lower abundance. Instead, the most prominent unprocessed species in control MEFs corresponded to a transcript spanning *16S-mtL1-mtNd1*, previously referred to as RNA19[49,50].

In the absence of mitochondrial methylation potential, the distribution of unprocessed transcripts shifted. Already at 8 weeks of age previously unidentified unprocessed reads accumulated in KO^SkM samples, and by 12 weeks, the majority (>50%) involved gene junctions within the ribosomal gene cluster or the *mtNd3-mtR-mtNd4/4L* region. This trend further increased in KO^MEFs, where these species accounted for over 70% of all unprocessed reads (Fig. 2b). To quantify the significance of this accumulation, we compared the abundance of polycistronic transcripts between control and KO samples relative to the total number of aligned reads (Fig. 2c). This confirmed a marked increase in unprocessed transcripts in mitoSAM-depleted samples. Notably, this increase was exclusively driven by the accumulation of unprocessed gene junctions involving tRNAs, while sites involving

non-canonical processing (e.g. *mtAtp8/6-mtCox3*, *mtNd5-mtCytb*, or 5'-leader-*mtCo1*) were not enriched and even declined in proportion to other unprocessed species (Fig. 2b, c and Supplementary Fig. 4). Among all regions, the ribosomal gene cluster (*mtF-12S-mtV-16S-mtL1*), previously referred to as RNA4[51], was most affected. Multiple processing intermediates originating from this region accounted for nearly 60% of unprocessed reads and close to 20% of total ONT reads in KO^MEFs (Fig. 2b, c).

Given that the depletion of mitoSAM predominantly affected sites of canonical processing, we next focused on polycistronic transcripts containing at least one mt-tRNA. These transcripts were already detectable in control samples, albeit at low levels (0.2% to 1.1%), seemingly accumulating slightly between 8- and 12-week-old quadriceps and MEFs (Fig. 3a). In *Samc* KO samples, however, this fraction increased substantially, reaching up to 15% of total mitochondrial reads in KO^MEF cells, correlating with the declining mitochondrial methylation potential. To identify the most affected transcript species, we plotted ONT sequencing data for unprocessed transcripts representing at least 0.01% of total reads (Fig. 3b), revealing three major

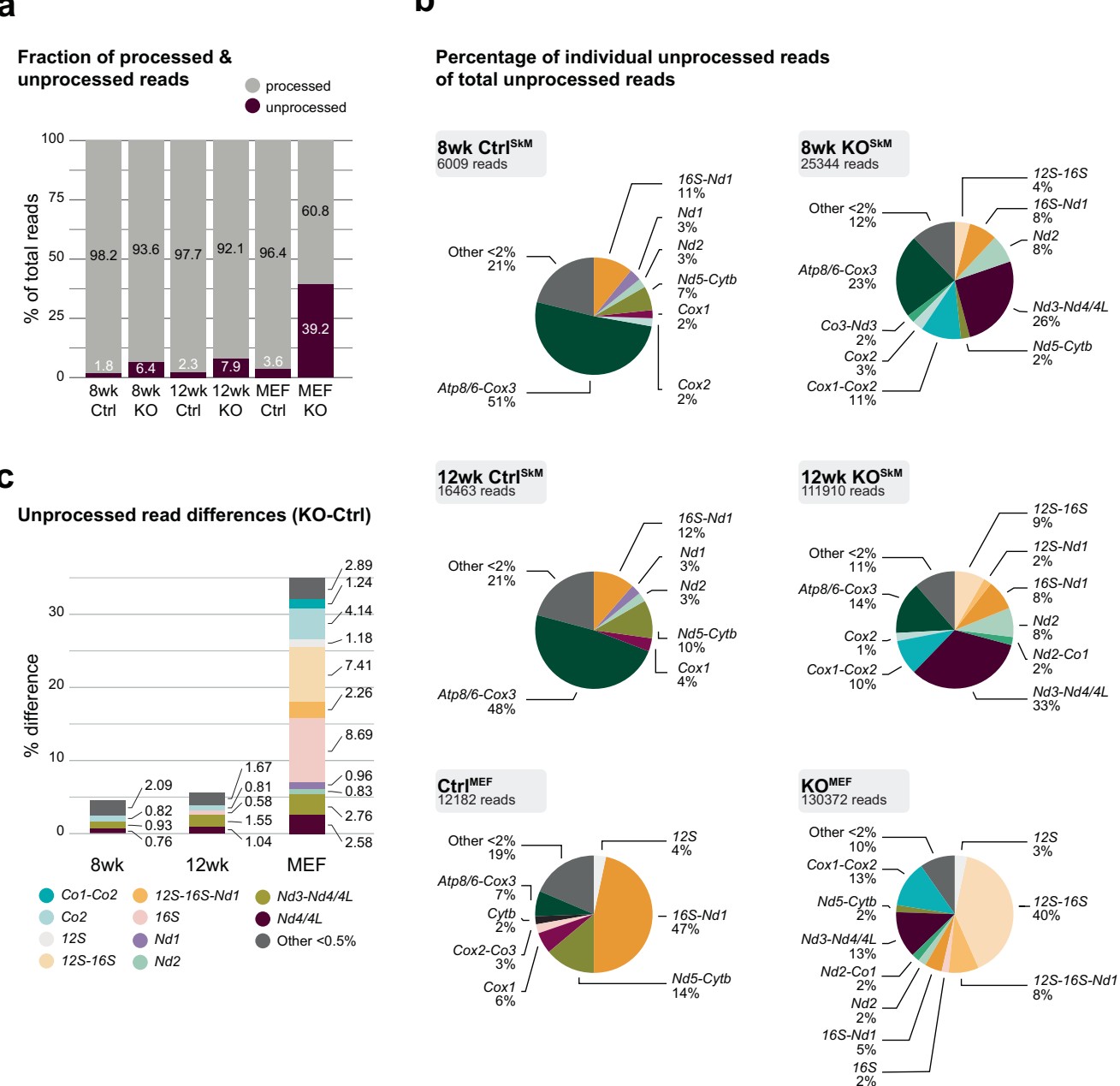

**Fig. 2 | ONT sequencing. a** Fraction of processed and unprocessed reads from total read counts in the indicated samples. Reads were classified as unprocessed if their ends did not fall entirely within ±20 nt of the annotated gene boundaries (see Supplementary Fig. 3a). **b** Proportions of individual unprocessed reads, containing mRNA and/or rRNA, as a percentage of the total unprocessed pool of reads for each sample. Reads representing less than 2% were pooled in the 'other' category. **c** Difference of fraction of individual unprocessed reads as a percentage of total reads in KO and Ctrl samples. Only junctions where the absolute difference exceeded 0.5% were visualised. Reads are indicated as most 5′ to 3′ mRNA or rRNA. Source data are provided as a Source Data file.

genomic regions of accumulation: (i) the ribosomal gene cluster, (ii) *mtNd2* along with flanking tRNAs spanning the entire *WANCY* tRNA cluster and (iii) *mtR* together with the *mtNd4*/4 L bicistron (Fig. 3b).

We specifically focused on the rRNA gene cluster of *mtF-12S-mtV-16S-mtL1*. Several tRNAs, including *mtF*, *mtL1* and *mtI*, retained either their 5′-leader or 3′-trailer sequences, deviating from the proposed strict 5′ to 3′ processing hierarchy[52]. Notably, *mtV* was never observed with only a 5′-leader sequence, suggesting that its 5′ cleavage is a critical step for processing this cluster.

To investigate this further, we mapped the start and end sites of all reads containing *12S*, *16S*, or *mtNd1* in KO^MEFs and KO^SkM quadriceps (Fig. 3c–e and Supplementary Fig. 5a). In KO^MEFs, for instance, only ~6% of *12S* and *16S* rRNA transcript, and ~40% of *mtNd1* transcripts, were

fully processed (Supplementary Fig. 5b). Interestingly, although ~90% of *12S* transcripts were fully released in control MEFs, the majority of *16S* rRNA transcripts (85%) remained incompletely processed, indicating a preferential release of *12S* over *16S* rRNA. We confirmed these processing intermediates by Northern blot analysis, where the rRNA gene cluster was the predominantly affected region (Fig. 3f and Supplementary Fig. 6). Comparing ONT sequencing coverage with qRT-PCR results highlighted an overrepresentation of *12S* and *16S* rRNA in KO^MEFs while increased *mtNd1* and decreased *mtCytb* steady-state levels were consistent between the two datasets (Fig. 1c and Supplementary Fig. 2). This discrepancy may reflect the polyA-enrichment of samples for ONT sequencing capturing rRNA attached to the more extensively polyadenylated *mtNd1*.

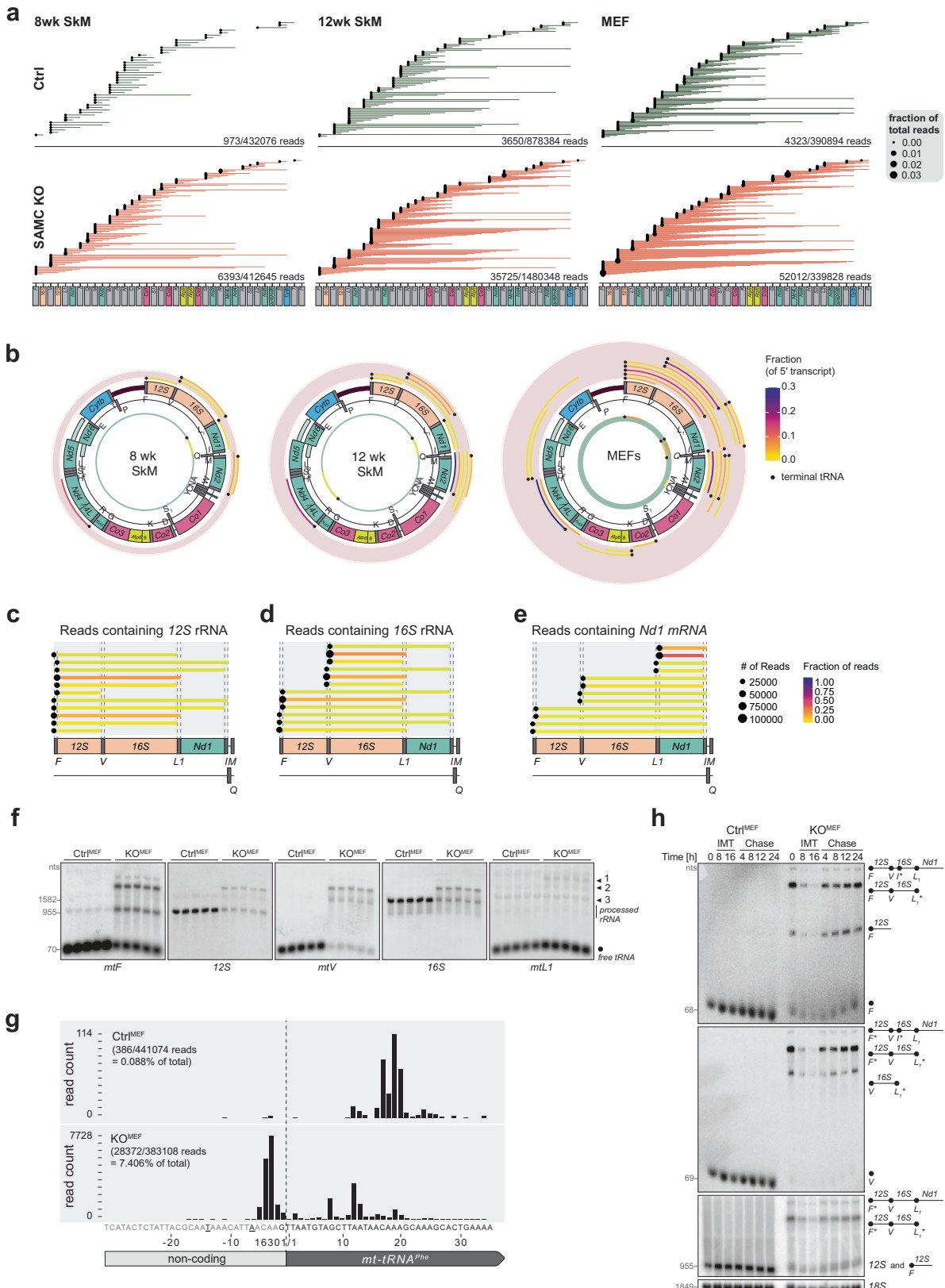

Due to technical limitations, ONT sequencing does not sequence the 5′ ends of transcripts, and the reads consistently mapped -12nt (± 2) downstream of the reported start sites (Supplementary Fig. 7). In agreement, in control samples, the 5′ ends of transcripts containing *mtF* mapped -20nt downstream of its annotated start site (Fig. 3g and Supplementary Fig. 8)[53]. In contrast, in KO[SkM] and KO[MEF] samples, most

start sites mapped further upstream, consistent with the reported heavy strand promoter (HSP) start site[53], thus demonstrating failed processing of the 5′ untranslated region of *mtF*.

To determine the stability of these unprocessed transcripts, we treated control and KO[MEFs] with an inhibitor of mitochondrial transcription (IMT1B)[54]. While the steady-state levels of fully processed

**Fig. 3 | Mitochondrial SAM is critical for processing of the rRNA gene cluster.**
**a** ONT sequencing data filtered for all reads containing tRNA sequences and mapped against the mitochondrial genome (KR020497). Control (Ctrl; green) and *Samc* KO (red) from 8 and 12-week-old muscle (quad) and MEF samples (*n* = 3 biologically independent samples). **b** ONT sequencing data filtered for reads containing tRNA sequences present in more than 0,01% of the total RNA pool and mapped against the mitochondrial genome (KR020497). Control (inner circle; green) and *Samc* KO (outer circle; pink) reads are shown from 8 and 12-week-old quadriceps and MEF samples. Black dots represent presence of a tRNA sequence at the 5′ or 3′ termini of the reads. Shown are the fractions of transcripts passing through at least the centre of one tRNA, calculated relative to reads in the next mRNA or rRNA. (*n* = 3 biologically independent samples). **c**–**e** Zoom-in of KO^MEFs RNA reads passing through (**b**) *12S* rRNA, (**c**) *16S* rRNA, or (**d**) *mtNd1* with no fraction cut-off. Gene borders are indicated by dotted lines. (*n* = 3 biologically independent samples). **f** Northern blot analysis of rRNA mitochondrial transcripts and flanking

tRNAs in control and *Samc* KO MEFs as indicated (*n* = 5 biologically independent samples). Nucleotide length provided in grey (nts). Black arrowheads indicate unprocessed transcripts (1) *mtF-12S-mtV–16S-mtL1-mtNd1* (2) *mtF-12S-mtV-16S*, (3) *mtV-16S*. **g** number and position of reads containing *mtF* (tRNA^Phe) in KO^MEFs is shown. Transcription start sites for the heavy strand promoter are indicated in bold. (n = 3 biologically independent samples). Number of reads and percentage of total reads is indicated. **h** Northern blot analysis of selected mitochondrial transcripts in control or *Samc* KO MEFs after treatment with 20 mM IMT1B, a mitochondrial transcription inhibitor, using probes against *mtF* (top panel), *mtV* (middle panel) or *12S* (bottom panel). Nucleotide length provided in grey (nts). (*indicates tRNAs for which the gel resolution does not allow us to distinguish whether they are present or absent.) 18S was used as loading control. Treatment times were 8 and 16 h, chase timepoints were at 4, 8, 12 and 24 hours. 0 = dimethyl sulfoxide (DMSO) vehicle control without IMT1B. Representative experiment of three independent experiments. Source data are provided as a Source Data file.

transcripts gradually decreased in IMT1B-treated cell lines (Fig. 3h), unprocessed intermediates exhibited a more rapid clearance, suggesting a faster turnover. Upon IMT1B removal, these species reappeared, implying that these rRNA processing intermediates respond dynamically to transcription rates. Thus, this data identify methylation within the rRNA gene cluster as critical for efficient processing within this region.

## Mitochondrial ribosome assembly depends on mitochondrial SAM

To investigate whether the altered mt-rRNA and mt-tRNA compositions affected translation, we examined the mitochondrial translation capacity in mitochondria isolated from quadriceps or MEFs. Mitochondrial translation was mildly reduced in 12-week-old KO^SkM samples, and no signal from nascent peptides was detected in KO^MEFs (Fig. 4a). To determine if this defect is specific to the altered mt-tRNA composition or monosome formation, we performed sucrose gradient sedimentation of mitochondrial extracts, followed by Western blot analysis for subunits of the mitoribosome. KO^SkM samples presented with a less distinct monosome peak and increased tailing, suggesting a mild assembly defect. A combination of abnormal monosome formation and modified tRNA composition probably causing the observed translation defect in KO quadriceps (Supplementary Fig. 9a). In agreement, KO^MEFs, which have a completely depleted mitoSAM pool, presented with reduced monosome formation signal from the mtSSU and mtLSU each shifted to the left by one fraction, consistent with lighter and/or aberrant assembly intermediates (Fig. 4b). Mass spectrometry-based label-free proteomics (Fig. 4c) and Western blot analysis (Fig. 4d) demonstrated that several factors of the mitochondrial translation machinery were markedly increased in KO^SkM quadriceps but remained unchanged or even reduced in KO^MEFs, consistent with a compensatory response in the muscle (Fig. 4c, d)[5,55]. Thus, while muscle can maintain nearly normal monosome formation and translation in an attempt to compensate for the OXPHOS defect, the sustained depletion of mitoSAM in KO^MEFs affects efficient monosome formation.

We next asked whether immature mtSSU and mtLSU were the cause of the failed monosome formation. To investigate this, we performed stable isotope labelling with amino acids in cell culture (SILAC) on MEFs, followed by separation of mitochondrial ribosomes on a sucrose gradient and mass spectrometry of fractions corresponding to the mtSSU, mtLSU and monosome to identify their respective compositions (Supplementary Fig. 9b–d, Supplementary Fig. 10a, b, and Supplementary Data 10). Generally, monosome formation requires an equal stoichiometry of mtSSU and mtLSU proteins. However, although we observed an overall decrease in ribosomal proteins in these fractions in the KO sample (Supplementary Fig. 10a), the distribution of mtSSU and mtLSU ribosomal proteins differed greatly between the fractions. Levels of mtSSU proteins increased between the mtSSU and

monosome fractions, whereas mtLSU proteins decreased in both the mtLSU and monosome fractions (Fig. 5a, b). Interestingly, five mtSSU proteins did not increase in the monosome fraction, of which uS11m, uS12m, bS21m and mS33 all have recently been suggested as 'late-binding MRPs'[56].

Surprisingly, we also observed a strong enrichment of several mitochondrial RNA binding proteins typically associated with processing and early stages of RNA maturation in the higher molecular weight fractions (15-17) of the KO^MEFs samples (Fig. 5c). For instance, TRMT10C was enriched in all three fractions, and several factors important during the early stages of the mtSSU, such as NOA1, ERAL1, and TFB1M, as well as the entire pseudouridylation (PUS) module, which is involved in *16S* rRNA modification, accumulated in the monosome fraction. We confirmed this distribution for RPUSD4 and ERAL1 on ribosome gradients using Western blot analysis (Supplementary Fig. 10c). Despite exhibiting a milder translation defect and normal mitoribosome sedimentation patterns, KO^SkM gradients also showed an enrichment of RPUSD4 in mtLSU and monosome fractions (Supplementary Fig. 9a), demonstrating the same mechanism for rRNA processing intermediate stabilisation in vivo.

Taken together, these results suggest that in the absence of methylation, mtSSU and mtLSU maturation arrests in immature states. Furthermore, the presence of early processing and maturation factors in gradient fractions typically associated with the monosome suggests the accumulation of a large RNA-protein complex undergoing processing and assembly simultaneously. This implies that mitoribosome assembly can begin before the complete processing of the mitochondrial ribosomal gene cluster.

## Processing of mt-tRNAPhe and mt-tRNAVal promotes early ribosome assembly

To test whether the unprocessed rRNA gene cluster is part of a larger complex, we isolated RNA from sucrose gradient fractions, followed by Northern blot analysis using probes against *mtF*, *mtV* and *mtL1*. In control MEFs, all three processed tRNAs sedimented with the free fractions (fractions 1-3). Interestingly, though, *mtF* and *mtV*, but not *mtL1*, also co-migrated with fractions corresponding to the mtLSU (fractions 10–12) and the monosome (fractions 13–15) (Fig. 5d). The mitochondrial ribosome differs from cytosolic and bacterial ribosomes in that, besides containing *12S* and *16S* rRNAs, it also includes a structural tRNA. In humans, this structural tRNA is *mtV*[57,58], while *mtF* has been reported in other mammals[59]. The presence of both *mtV* and *mtF* in the monosome fraction of control MEFs suggests that both tRNAs act as structural components of the murine mitochondrial ribosome. In KO^MEFs, the structural tRNA shifted towards *mtF*, probably due to the low levels of *mtV*. Furthermore, in KO^MEFs, both structural tRNAs, but not *mtL1*, also accumulated as unprocessed transcripts in the higher molecular weight fractions (Fig. 5d), suggesting they are part of a separate, larger protein-RNA complex. Thus, our results

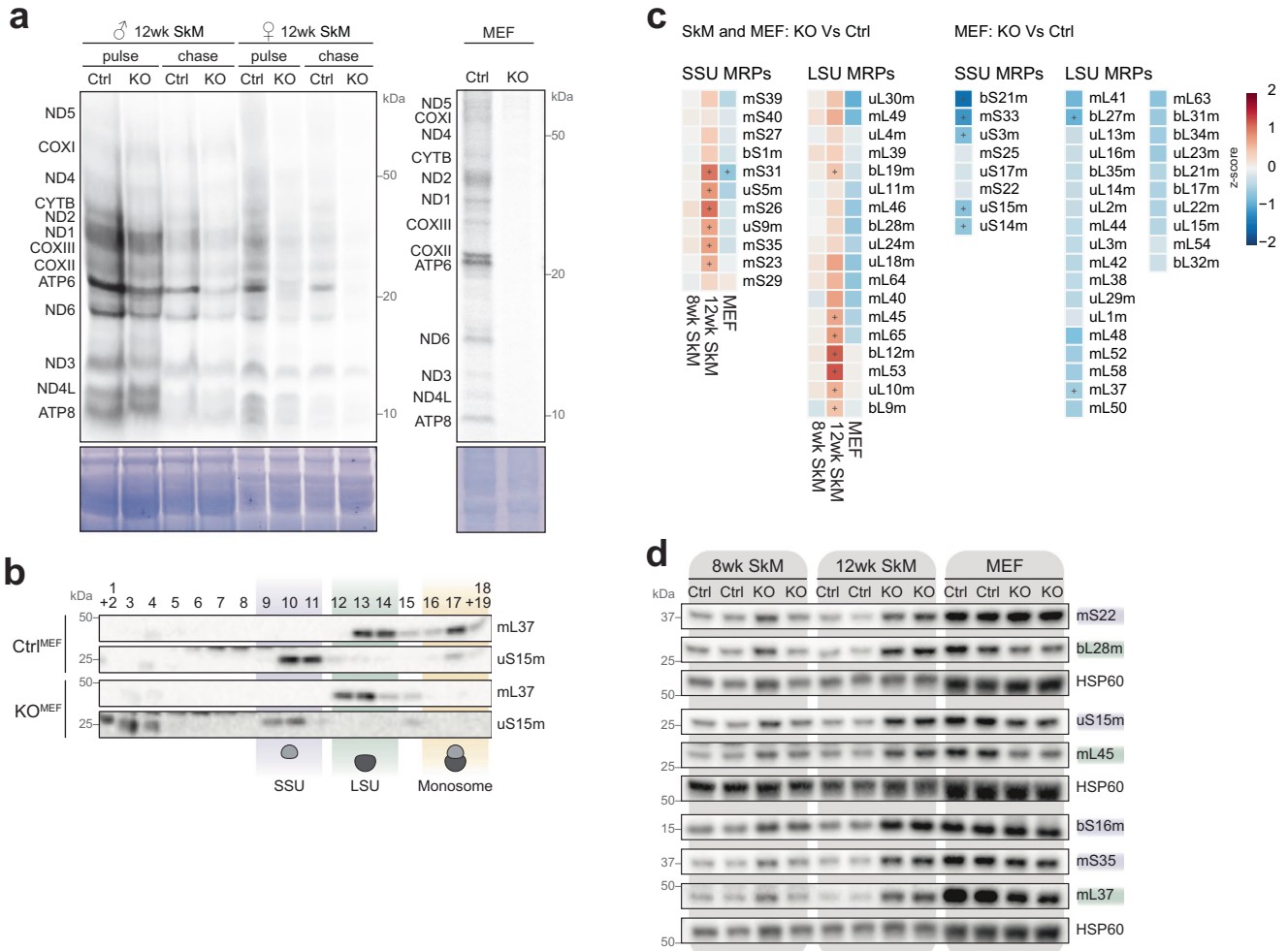

**Fig. 4 | Mitochondrial ribosome assembly depends on SAM. a** De novo translation in isolated mitochondria from male and female 8 week and 12-week-old muscle or MEFs samples as indicated. Expected mitochondrial proteins are shown. Coomassie stain of the gel indicates loading. **b** Western blot analysis of ribosome gradient fractions from *Samc* KO and control MEFs, probed with antibodies against the small (uS15m) and large (mL37) mitochondrial ribosome subunits. The small (28S), large (39S) and monosome (55S) fractions are indicated. A representative experiment is shown of three independent experiments performed with independently prepared samples. Nearest molecular weight indicator provided in grey (kDa). **c** Proteomic levels of mitoribosome subunits from 8 to 12-week-old muscle or MEF *Samc* KO samples normalised to control samples. (*n* = 3). **d** Western blot analysis of mitochondrial ribosome subunits in 25 µg of mitochondrial lysate from 8 to 12-week-old muscle or MEF samples. Nearest molecular weight indicator provided in grey (kDa). Source data are provided as a Source Data file.

indicate that *12S* and *16S* rRNAs initiate early assembly stages but stall due to incomplete rRNA processing in the absence of mitoSAM.

## mtLSU assembly requires mitoSAM

The observed stalling of rRNA cleavage exposed a critical requirement for mitoSAM to facilitate entry into the mitoribosome assembly process. Additionally, complete monosome formation was impeded in the absence of mitochondrial methylation potential. We used cryogenic electron microscopy (cryo-EM) on mitochondrial preparations from *Samc* KO[MEFs] to visualise these assembly intermediates directly. Reconstructions from the cryo-EM dataset, followed by 3D classification and refinement, identified ten distinct classes representing various stages of mtLSU maturation, and no mature mitoribosomes, allowing us to establish an assembly sequence (Fig. 6a and Supplementary Fig. 11–13). States A1-A4 are the earliest mtLSU assembly states identified here, characterised by an immature central protuberance (CP) and an A-loop of H92 held by an MRM3 dimer, which would typically methylate G2482 (Fig. 6a). These states exhibit a relatively low overall resolution (4.6–7.4 Å) which is sufficient to trace CP maturation and the dissociation/recruitment of assembly factors and mitoribosomal proteins.

States B-D exhibit a mature CP but still an immature PTC, which remains bound by NSUN4-MTERF4 heterodimer along with an array of state-specific factors, including GTPases, GTPBP7 (states B-D), GTPBP10 (states B1,2) and the methyl transferase MRM2 (state D). States B1-C1 were captured by incubating mitochondrial lysate with GMPPNP and subjecting the sample to chemical cross-linking prior to grid freezing. Consequently, GTPBP7 adopts its GMPPNP-bound pre-hydrolysis conformation as observed previously[60,61] and GTPBP10 could be stabilised in states B1,2. The states B-D, with an overall resolution ranging from 2.8 to 3.62 Å, allowed the identification of bound factors and conformational changes in rRNA and protein elements at the maturing PTC (Supplementary Fig. 11–13). Finally, particles from states B-D, which exhibited a mature CP were pooled to generate a CP map at 3.0 Å resolution, enabling the characterisation of the structural tRNA (Fig. 6b and Supplementary Fig. 11 and 12).

Together, the multiple assembly states spanned from those with an immature CP and disordered PTC to those in which *16S* rRNA H80, H89 and H92 had adopted a near-mature conformation, despite lacking methylation (Fig. 6b). However, the PTC remains immature, preventing full mtLSU maturation and monosome formation (Fig. 7).

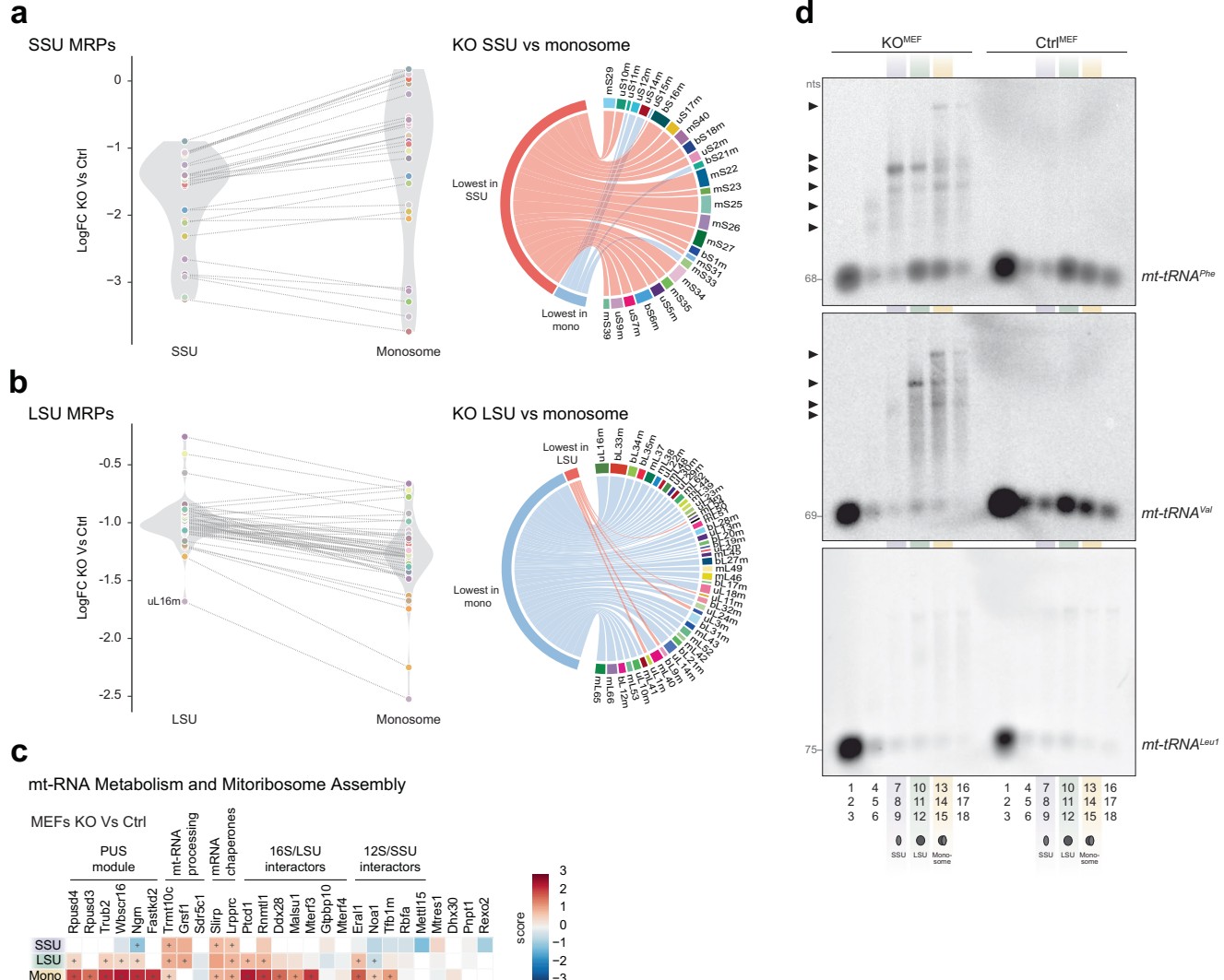

**Fig. 5 | The unprocessed mitochondrial rRNA cluster engage in early ribosome assembly.** Distribution of subunit levels of the (**a**) small or (**b**) large mitochondrial ribosome in fractions representing the mtSSU or monosome after SILAC labelling of *Samc* KO and control MEFs ($n = 2$). Right panel shows changes of individual subunits that are either increased (red) or decreased (blue) in the monosome fraction. Right panel Log fold change, and distribution of (**a**) mtSSU (**b**) mtLSU factors in the respective fractions. **c** Enrichment levels of mitochondrial RNA binding proteins and ribosome assembly proteins in ribosome gradient fractions in *Samc* KO and control MEFs after SILAC labelling ($n = 2$). **d** Northen blot analysis of ribosome gradient fractions from *Samc* KO and control MEFs ($n = 1$). Nucleotide length provided in grey (nts). Source data are provided as a Source Data file.

The earliest previously reported assembly state is characterised by an immature CP bound by DDX28, an MRM3 dimer stabilising H92, and the presence of the ObgE-like GTPase (GTPBP10) that stabilises H89 in an immature conformation[62–64]. Here, we observed four sub-states (A1-A4) with bound MRM3 dimer, which differ in (i) binding of uL16m (as A1/A3 mature to A2/A4) and bL36m (as A1/A2 mature into A3/A4) at the base of the L12 stalk and (ii) maturation status of the CP coupled with the dissociation of DDX28 (as A1/A2 mature to A3/A4) (Fig. 6a and Supplementary Fig. 14a). Notably, GTPBP10 did not yet appear in the assembly pathway reported here. However, in states A2 and A4, the conformation of H89 correlated with the occupancy of uL16m, where H89 is lodged against uL16m proximal to a basic patch on its surface. It is thus likely that in the absence of uL16m (states A1/A3), H89 becomes disordered or displaced (Supplementary Fig. 14a). Interestingly, uL16m was strongly reduced in the KO^MEFs mtLSU and monosome fractions in our SILAC gradient analysis, assigning uL16m a critical role during mtLSU assembly. In states A3/A4, DDX28 dissociated, allowing the recruitment of bL33m and the adoption of a more mature and structured CP. The MRM3 dimer remained bound to

H92 in these states, but the weak density suggested that this was only a loose interaction (Fig. 6a).

Interestingly, despite the compromised processing of *mtF* and *mtV*, the CP contained a structural tRNA. The CP-tRNA density (with a local resolution ranging from 2.99 to 5.5 Å) likely represents a mixture of *mtF* and *mtV*, with higher proportion of *mtF* (Fig. 6b). There was no clear density for methylations at A9 and G10 (local resolution ~3.1 Å) or for 3′CCA (local resolution ~4.9 Å at the terminal residue of *mtF* and ~4.0–4.5 Å at the adjacent mL46 loops) indicating that the structural tRNA potentially lacks a 3′CCA and is not amino-acylated (Fig. 6b and Supplementary Fig. 12b). Consistently, no density for 3′CCA RNA modification was observed in any mature tRNAs suggesting that this modification is not required for incorporation into the mitochondrial ribosome.

In state B1, the CP was completely mature and dissociation of MRM3 dimer was accompanied by reorganised rRNA helices. Specifically, we observed (i) binding of the NSUN4-MTERF4 complex, which sequesters H68-71, (ii) recruitment of GTPBP7 and GTPBP10, (iii) the absence of uL16m, (iv) the presence of bL36m, albeit at partial

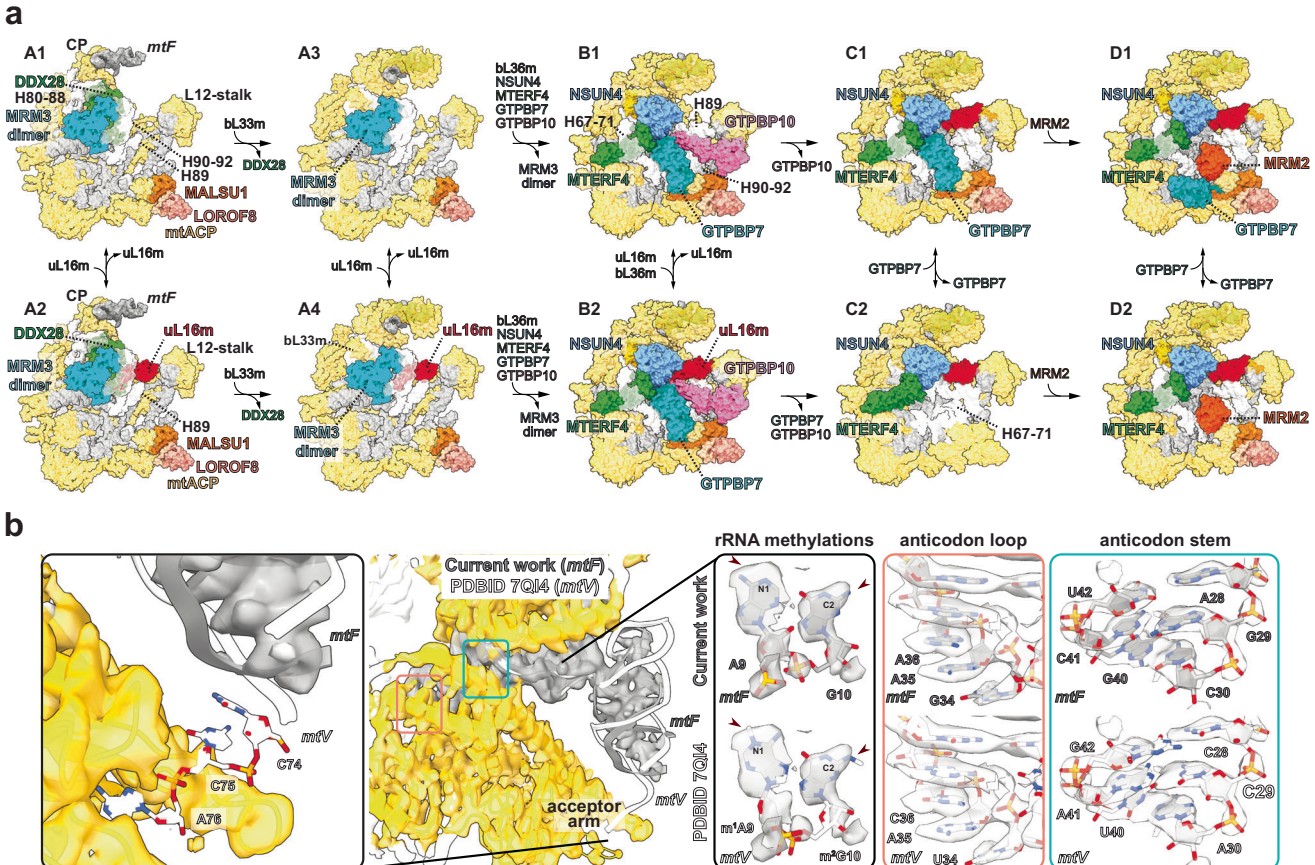

**Fig. 6 | Cryo-EM structures of mouse mitochondrial LSU assembly. a** Overview of identified mtLSU states (A-D). *16S* rRNA helices in white, CP-tRNA in grey, mature structures in yellow. Large ribosomal subunit protein uL16m (red). Biogenesis factors shown are highlighted as followed: rRNA methyltransferase 3, MRM3 dimer (cyan); DEAD box RNA helicase 28, DDX28 (green); GTP-binding protein 10, GTPBP10 (magenta); mitochondrial assembly of ribosomal large subunit protein 1, MALSU1 module (orange, salmon, pink); transcription termination factor 4,

MTERF4 (dark green); 5-methylcytosine rRNA methyltransferase, NSUN4 (cyan); GTP-binding protein 7 GTPBP7 (turquoise); rRNA methyltransferase 2, MRM2 (dark orange). **b** Close up of the CP showing, *mtF* modelled into the density map and superposed with *mtV* (PDB ID 7QI4). Zoom-in panels show a lack of 3′ CCA (left), lack of rRNA methylations at positions 9 and 10, comparison of anticodon loop and part of the stem of *mtF* and *mtV* against the density map.

occupancy, and (v) an ordered H89 that is displaced by ~25 Å to a distinct conformation stabilised by GTPBP10 (Fig. 7a and Supplementary Fig. 14b). This state differed from a previous report due to the absence of uL16m (PDBID 8PK0[61]) which would potentially clash with H89 in this observed conformation (Fig. 7a and Supplementary Fig. 14b). In state B2, uL16m was stably accommodated, and H89 moved ~14 Å to align with a basic cleft formed by uL16m and GTPBP10, acquiring its mature position. This provides insights into how these two factors promote H89 folding (Fig. 7a and Supplementary Fig. 14b). As GTPBP10 dissociated from the maturing mtLSU, GTPBP7 remained bound in its pre-hydrolysis conformation (state C1; Fig. 7a), as previously reported[60]. Finally, states D1/D2 represented the most mature states identified and characterised by the recruitment of MRM2, as GTPBP7 adopted a less ordered flipped-out conformation (Fig. 7a)[65].

## MitoSAM is required for PTC maturation
During translation, peptide bond formation occurs in the PTC, predominantly formed by *16S* rRNA during the late stages of mtLSU assembly and involves the folding and methylation of the A and P loops[34,60,62,63,65–67]. The absence of mitoSAM did not prevent the recruitment of assembly factors in the ten mtLSU states reported here. Rather, we observed a partially disordered PTC. In the most mature assembly state identified here (state D), H68-71 remained sequestered by NSUN4-MTERF4, while MRM2 was retained in its canonical position, holding the A-loop in its catalytic pocket to methylate U2481 (U3031;

human mitochondrial genome numbering) (state D1 in Fig. 7a). GTPBP7 was flipped-out and largely disordered while the PTC loops between H74-H89 and H89-H90 were only partly ordered (Fig. 7a, b).

According to the canonical assembly pathway, state D is followed by the recruitment of GTPBP5, promoting a significant remodelling of the PTC loops and the release of MRM2 from the A-loop post-methylation[66,68,69]. The lack of GTPBP5 binding in our model can be attributed to (i) an absence of A-loop methylations at U2481 and G2482 (U3039 and G3040; human mitochondrial genome numbering); (ii) a clash between the unreleased MRM2 (due to the unmethylated A-loop) and incoming GTPBP5; and (iii) an increased disorder of the PTC loops. Thus, in the absence of mitoSAM, multiple causes result in an unstable interface, preventing the binding of the Obg domain loops of GTPBP5.

It is not known when MRM1 methylates the P-loop during *16S* rRNA maturation and mtLSU assembly, but the methylation could already be observed in states containing GTPBP5 and GTPBP6[66,69]. How MRM1-dependent methylation affects PTC remodelling is also unclear, but direct contact between Gm2253 (Gm2815 in humans) and the PTC loops (H89-H90; H74-H89) has been noted in human assembly intermediates (PDBID 7OF7[66]) and mature mtLSU (PDBID 7QI4[24]) (Fig. 7b). The increased disorder of PTC loops could therefore be partly due to a lack of Gm2253 methylation. The short segment of PTC loops that is ordered is likely scaffolded by methyltransferases MRM2 and NSUN4 (Fig. 7b) Overall, our results suggest that the P-loop and the proximal PTC loop are distorted without methylation, inhibiting PTC formation.

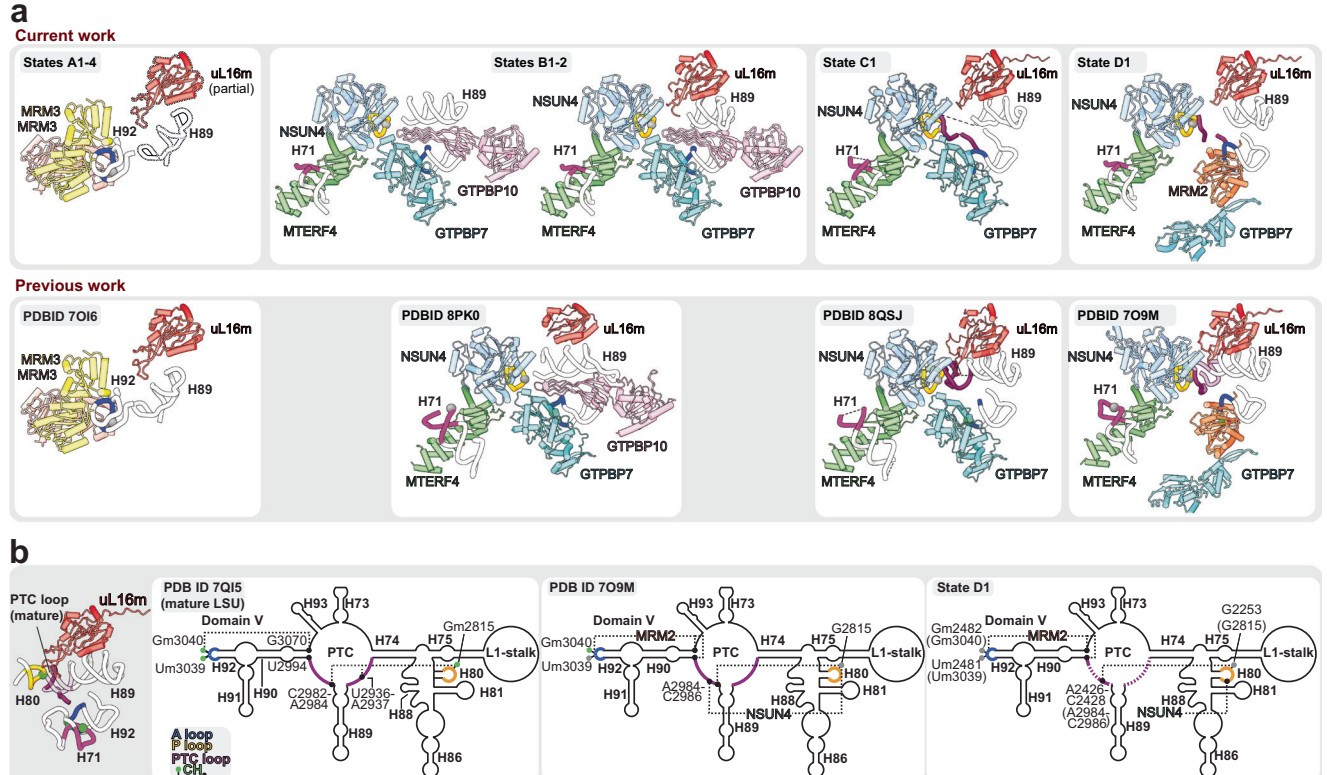

**Fig. 7 | Formation of the PTC is inhibited in Samc KO MEFs. a** Close-up view of PTC maturation states. Panels show mtLSU assembly factors bound to part of rRNA helices and loops, contributing to the PTC, namely, h67-71 (pink), h80/P loop (gold), PTC loops (purple) and h92/A loop (blue). Lower panels show comparable states from published work while upper panels depict states A1-4, states B1, B2, C1 and D1 to depict the key stages of PTC maturation observed in current work. **b** 2D representation of domain V in mature mtLSU, human mtLSU intermediate (PDB ID 7O9M) and state D1. Interactions made by methyl transferases MRM2 and NSUN4 are highlighted. Residues methylated in mature mtLSU are highlighted (methylated as light green; unmethylated as grey). Dashed lines represent predicted stabilising interactions between methylated residues. Residue numbers are shown according to mouse genome numbering and the corresponding human numbering is indicated in brackets.

## Discussion

Methylation plays a crucial role in processing mitoribosomal RNAs and mtLSU assembly and is therefore essential for mitochondrial gene expression. Mitochondrial rRNAs and tRNAs carry various post-transcriptional modifications, and although several have been linked to stabilising the decoding centre, their broader roles in function, stability, assembly, or regulation remain unclear[24]. Here, we selectively targeted the mitochondrial methylation potential to investigate SAM-dependent modifications within mitochondria in both mouse and MEFs, without directly affecting the responsible methyltransferases. Our findings establish methylation as a crucial determinant of mitochondrial translation by ensuring tRNA stability, proper mt-rRNA processing, and mitoribosome assembly.

Recent work by the Churchman lab demonstrated that processing of the rRNA gene cluster is rate-limiting for mitochondrial translation and necessary for coordinating nuclear and mitochondrial translation[47]. Our findings expand this model by showing that methylation is essential for this process, particularly in the processing of *mtV*. This places TRMT10C-mediated methylation—and by extension mito-SAM—at the centre of efficient mitochondrial gene expression. This contradicts several previous reports that suggested that mt-tRNA methylation by MRPP3 was dispensable for RNase P activity[10,13,14]. Reducing the mitochondrial methylation potential in vivo and in cell culture resulted in the accumulation of several unprocessed transcripts involving the rRNA gene cluster, *mtNd2* and *mtR-mtNd4/4 L*. The specific sensitivity of these regions remains unclear, but in the case of the rRNA gene cluster, the extensive secondary structures of the rRNAs may contribute. Structural reconstructions with these leader and trailer sequences will be essential to clarify whether these junctions fold differently. Furthermore, we identified polycistronic transcripts with tRNAs at both their 5′ and 3′ termini, challenging the hierarchical cleavage of canonical gene junctions.

Interestingly, non-canonical processing, i.e. gene junctions not containing tRNAs, appeared largely unaffected by the reduced methylation potential. These sites are processed independently of RNase P and ELAC2 and rely on members of the FASTK family and the 2′,3′ cyclic phosphatase, ANGEL2[11,44,46]. Our ONT sequencing analysis of mouse quadriceps suggests that even in control samples, non-canonical processing is inherently less efficient than canonical, with the most abundant unprocessed transcript being the previously described tricistronic *mtAtp8/6-mtCo3* transcript. Since the flanking canonical junctions were efficiently cleaved, it is tempting to speculate that non-canonical cleavage proceeds more slowly or at a different intramitochondrial location. A microdeletion in the *mtAtp8/6-mtCo3* gene junction does not impair its processing, suggesting that structural constraints are not limiting[70]. Whether this tricistron has a regulatory role remains unclear, although it has been proposed to be efficiently loaded onto the mitoribosome and translated[47,48].

Our work demonstrates that a decreased mitochondrial methylation potential restricts mitochondrial gene expression by impeding the processing of the ribosomal gene cluster. This, in turn, results in incomplete ribosome assembly due to the lack of *mtF* and *mtV* release from *12S* and *16S* rRNA. In addition, we show that the 2′-O-methylations of *16S* rRNA are crucial for proper folding of the PTC and further maturation of the mtLSU. This contrasts with recent data from human HEK 293 T cells, which proposed that the 2′-O-methylation of U2481

(U3039 in human mtDNA numbering) is dispensable for monosome formation[34]. Supporting our results, a dependency on methylation has recently been demonstrated for the cytosolic LSU in yeast, where a single 2′-O-methylation on the pre-60S rRNA subunit was shown to be essential for ribosome assembly[71].

Our cryo-EM analysis of the mtLSU from SAM-depleted mitochondria reveals ten distinct assembly intermediates demonstrating how methylation can orchestrate mitoribosome maturation. Early states (A1–A4) exhibit an immature CP with MRM3 dimers bound to H92, poised to methylate G2482. Later states, (B–D) retain NSUN4-MTERF4, GTPBP7/10 and MRM2 but in the absence of *16S* rRNA methylations (U2482/G2482) fail to stabilise the PTC. Consequently, H89 and H92 are immature, and PTC loops become disordered, blocking *GTPBP5* recruitment and monosome formation.

The mitoribosome assembly intermediates reported here share similarities with *E. coli* ribosome assembly. In *E. coli*, depletion of SAM or the methyltransferase RlmE (*E. coli* homologue of mitochondrial MRM2) also leads to accumulation of LSU precursors with defective PTC formation, and delayed assembly[31,72]. Like MRM2, *RlmE* methylates 23S rRNA at U2552 (U2481 in mouse), and loss of the methylation destabilises helix H89–H92 interactions that are critical for PTC folding[32]. In addition, uL16m and bL36m are absent from the accumulated intermediate[33]. Bacterial LSU assembly intermediates bound by the ObgE GTPase (homologue of mitochondrial GTPBP5 and GTPBP10) exhibit incomplete maturation of homologous rRNA helices (H90–93)[64]. Since in mitochondria there are two ObgE homologues, GTPBP5 and GTPBP10, rRNA H89 can be correctly accommodated and uL16m and bL36m are stably recruited by GTPBP10, as illustrated in our B2 state. This is consistent with cryo-EM structures of bacterial LSU assembly intermediates, further supporting a conserved role for ObgE in ribosome biogenesis and adding to the previously reported human mtLSU intermediate (PDB 8PK0[61]). Furthermore, SAM-depletion induced assembly defect in *E. coli* could be partially rescued by overexpression of *RlmE*[72] and its catalytic activity was found to be necessary[32] which is in line with our observation that MRM2 bound state is the most mature intermediate accumulated. Overall, these conserved defects highlight the universal role of *rRNA* methylation in stabilising tertiary interactions during LSU biogenesis, while mitochondria have evolved additional checkpoints, such as GTPBP10 and the NSUN4-MTERF4 sequestration of H68–71, to regulate this process.

Another key finding from our cryo-EM analysis relates to the incorporation of structural *tRNA* (*mtV* or *mtF*) into the *CP*, a defining feature of the mitoribosome[57,58]. Previous studies have shown that mtV is universally present alongside *12S* and *16S* rRNA, while *mtF* can substitute for *mtV* in some species and even under disease conditions[59]. At a local resolution of 4.9 Å, we observed a mixture of *mtF* and *mtV* incorporated within the *CP*, yet lacking the canonical 3′ CCA modification. This suggests that these structural tRNAs are integrated early during mtLSU assembly, prior to final maturation. Complementary SILAC and gradient analyses further support this model, revealing stalled ribosome maturation and an enrichment of processing factors (e.g. TRMT10C, ERAL1) in aberrant ribosome fractions. The absence of the 3′ CCA addition implies that this modification is not required for CP incorporation, and it further suggests that the mitoribosomal gene cluster may be processed and matured separately from other mitochondrial transcripts. Whether the interchangeable presence of *mtV* and *mtF* serves a regulatory function or provides an adaptive advantage remains to be explored.

We previously demonstrated that the mitochondrial SAM pool reflects its cytosolic production, providing a direct opportunity to regulate mitochondrial translation via one-carbon metabolism[5]. Here, we suggest a crucial role for mitoSAM in gating early mitochondrial ribosome biogenesis and mtLSU maturation. This, together with the recent observation that NSUN3-mediated methylation of *mtM* can drive mitochondrial translation[73], establishes a central role for the one-carbon cycle in controlling mitochondrial translation.

## Methods

### Mouse husbandry and tissue collection
*Slc25a26* conditional KO mice (*Samc^{loxP/loxP}*) were generated previously by Taconic Biosciences (Germany), by flanking exon 3 of the National Center for Biotechnology Information (NCBI) transcript NM_026255.5 with loxP sites[5]. *Samc^{loxP/loxP}* mice were crossed to *Mlc1f-Cre* mice[74] to generate skeletal muscle-specific *Slc25a26* KO (*Samc* KO^{SkM}) mice. All mice were maintained on a C57BL/6 N genetic background and kept in individually ventilated cages at ambient room temperature of 22–24 °C with a 12 h:12 h light:dark cycle and *ad libitum* access to food (Special Diet Services) and water. Groups included male and female animals unless indicated. Animal studies were approved by the local animal welfare ethics committee (Stockholm ethical committee) and performed in compliance with national and European law.

### Generation of MEF cell lines and cell culture
MEFs were previously derived from embryonic day 13.5 embryos from intercrossed *Slc25a26^{loxP/loxP}* mice. In brief, isolated *Slc25a26^{loxP/loxP}* MEFs were cultured at 37 °C and 5% CO$_2$ in medium comprising of Dulbecco's Modified Eagle's Medium (DMEM), high glucose and GlutaMAX (Thermo Fisher Scientific), supplemented with 10% foetal bovine serum (ThermoFisher Scientific), and 1% penicillin/streptomycin (ThermoFisher Scientific). Immortalised *Slc25a26^{−/−}* cells were obtained by transiently expressing Cre-recombinase. Transfected cells were serially diluted to separate single cells, and subsequent clones were screened by PCR analysis. MEF cells were cultured in high glucose Dulbecco's Modified Eagle Medium (DMEM) GlutaMax™, supplemented with 50 μg/mL Uridine (Sigma Aldrich), 10% Foetal Bovine Serum (ThermoFisher Scientific) and 1% Penicillin/Streptomycin (ThermoFisher Scientific). Incubator conditions were set to 37 °C and 5% CO$_2$. No commonly misidentified cell lines were used in this study.

### Mitochondrial enrichment from mouse tissue and cell lines
For enrichment of mitochondria from mouse quadricep, fresh or snap-frozen tissue was chopped until paste-like and transferred into a tissue mitochondrial isolation buffer (MIB-T) containing 225 mM Sucrose, 20 mM Tris, 1 mM EGTA. Trypsin was added at a 0.4% working concentration and samples were rotated at 4 °C for 10 min. Samples were diluted with MIB-T containing 0.5% BSA and 1x Protease Inhibitor (Roche) prior to homogenisation on ice with 25 strokes on a Schuett homgen^{plus} set to 800 rpm. Tissue homogenates were differentially centrifuged at 1000 × *g* and 11,000 × *g* and the resulting mitochondrial pellets were re-suspended in MIB-T containing 0.4 mg/mL Trypsin Inhibitor before a final spin at 11,000 × *g*.

MEF cells were scraped from cell culture vessels and resuspended in ice cold Dulbecco's Phosphate Buffered Saline then pelleted with a 5 min 800 × *g* spin at 4 °C. The pellet was resuspended in a cell mitochondrial isolation buffer (MIB-C) containing 150 mM D-mannitol, 100 mM Tris pH7.4, 1 mM EDTA with added BSA at a working concentration of 0.1%. Cells were then homogenised on ice with 20 strokes on a Schuett homgen^{plus} set to 600 rpm. Cell homogenates were differentially centrifuged at 1000 × *g* and 11,000 × *g*. The resulting mitochondrial pellets were resuspended in MIB-C without BSA before pelleting at 11,000 × *g*.

Mitochondrial pellets from tissue and cells were either quantified for immediate experimental use or resuspended in a freezing buffer containing 300 mM Trehalose, 10 mM Tris-HCl pH7.4, 10 mM KCl, 1 mM EDTA, 0.1% BSA for long-term storage at −80 °C.

### Western blot analysis
Mitochondrial pellets from MEF cells and quadricep were lysed in a mitochondrial lysis buffer (MLB) containing 50 mM KCl, 20 mM MgCl$_2$,

10 mM Tris pH 7.5, 1× Protease Inhibitor (Roche) and 1% Triton X-100. Mitochondrial lysates were combined with NuPAGE LDS sample buffer, loaded onto 4-12% NuPAGE Bis-Tris gels (ThermoFisher Scientific) for electrophoresis in an XCell SureLock system (ThermoFisher Scientific) and subsequently transferred to PVDF membranes using the iBlot 2 Dry Blotting System (ThermoFirsher Scientific). Membranes were blocked in 5% (w/v) milk in PBS with 1% (v/v) Tween 20 (PBS-T; Sigma-Aldrich). Membranes we incubated in primary antibody overnight at 4 °C and secondary antibody for one hour at room temperature, then developed using Clarity Western ECL substrate (Bio-Rad). All uncropped and unprocessed scans of western blots, together with size indication ladders, can be found in the main figure and supplementary source data files. A list of antibody suppliers, catalogue numbers and working dilutions can be found in Supplementary Data 11.

### Sucrose gradient sedimentation
Mitochondria were lysed in MLB with freshly supplemented RNase Block Ribonuclease Inhibitor (Agilent). Linear 10-30% sucrose gradients in 1x sucrose gradient buffer (50 mM KCl, 20 mM MgCl2, 10 mM Tris pH 7.5, 1× Protease Inhibitor) were generated in 14 × 89 mm Ultra-Clear centrifuge tubes (Beckman Coulter), using a 107 Gradient Master (BioComp), followed by ultracentrifugation for 15 h at 79,000 × $g$ at 4 °C in an Optima L-80 XP Ultracentrifuge (Beckman Coulter SW41-Ti rotor). Starting from the top of the gradient, 25 × 450 µL fractions were collected. Fractions for western blot analysis subsequently underwent Trichloroacetic Acid (TCA) precipitation. 1/100th volume of 2% Sodium Deoxycholate was added to each fraction followed by 20 minutes incubation on ice. 1/4th volume of 72% TCA was added and fractions were spun at max speed in a benchtop centrifuge for 20 minutes at 4 °C. Each fraction then underwent 2 washes in 100% acetone and resulting pellets were left to air dry under a fume hood. Precipitates were resuspended in NuPAGE LDS Sample Buffer freshly supplemented with 10 mM of Dithiothreitol (DTT) and either loaded directly onto an SDS-PAGE gel or frozen at −20 °C for short term storage.

### *In cellulo* and *in organello* mitochondrial translation assays
MEF were seeded into 6 well plates and cultured for 48 h in high glucose Dulbecco's Modified Eagle Medium (DMEM) GlutaMax™, supplemented with 50 µg/mL Uridine (Sigma Aldrich), 10% Foetal Bovine Serum (ThermoFisher Scientific) and 1% Penicillin/Streptomycin (ThermoFisher Scientific). Two 5 min washes with 2 mL of Cys-/Met-free medium containing DMEM High Glucose, No Glutamine, No Methionine, No Cysteine (ThermoFisher Scientific) supplemented with 10% Dialysed Foetal Bovine Serum, 1× GlutaMax, 1x Sodium Pyruvate (ThermoFisher Scientific) and 100 µg/ml Emetine. Fresh Cys-/Met-free medium was added, and the cells were incubated for a further 20 min. 200 µCi of [35S]-methionine/cysteine EasyTag Express protein labelling mix (Perkin-Elmer) was added to 1 mL of Cys-/Met-free medium per well and cells were incubated for 1 h at 37 °C and 5% CO$_2$. Following labelling, cells were washed three times with PBS, then harvested using cell scrapers and pelleted with a 5 min, 5000 × $g$ spin at 4 °C. Cell pellets were resuspended in PBS supplemented with 1x Protease Inhibitor (Roche) and 50U of Benzonase (Sigma Aldrich) and then frozen overnight at −20 °C to enable freeze-thaw lysis. Protein content of lysates was determined with a Pierce BCA assay (ThermoFisher Scientific), according to the manufacturers protocol, and mixed with NuPAGE LDS sample buffer, ready for SDS-PAGE.

Mitochondria from tissue quadricep were enriched through differential centrifugation in MIB-T. 500 ug of mitochondria were resuspended in 750 µg of translation Buffer (100 mM Mannitol, 10 mM Sodium Succinate Dibasic Hexahydrate, 80 mM Potassium Chloride, 5 mM Sodium Chloride, 1 mM Potassium Phosphate Dibasic, 25 mM Hepes, 60 µg/mL 17 Amino Acid Mix (Ala, Arg, Asp, Asn, Glu Gln, Gly, His, Ile, Leu, Lys, Phe, Pro, Ser, Thr, Trp, Val), 60 µg/mL Cysteine, 60 µg/mL Tyrosine, 5 mM ATP, 200 µM GTP, 6 mM Creatine Phosphate,

60 µg/mL Creatine Kinase and 200 µg/mL Emetine) with 150 µCi of EasyTag L-[35S]-Methionine. Samples were incubated for a 1-h pulse labelling, then chase samples were washed with isotope-free Translation Buffer, supplemented with 60 µg/mL cold methionine, and incubated for a further 3 h. Samples were washed one final time prior to resuspension in NuPAGE LDS sample buffer.

Samples from *in cellulo* or *in organello* labelling were loaded onto a 12% NuPAGE Bis-Tris gel (ThermoFisher Scientific) and electrophoresed at 150 V in an XCell SureLock system (ThermoFisher Scientific) for approximately 1 h. The gel was incubated in Imperial Protein Stain (ThermoFisher Scientific) and imaged to assess loading, prior to fixing in 20% Methanol, 7% Acetic acid, 3% Glycerol for 1 h at RT. After fixation the gel was dried under vacuum at 60 °C for 2 h, then placed to expose in a Fuji Film Phosphor Screen cassette. The resulting signal was detected using the Typhoon FLA 7000 Phosphorimager (GE Healthcare).

### RNA isolation, reverse transcription and quantitative PCR
RNA was isolated using RNeasy Fibrous Tissue Mini Kit (Qiagen), RNeasy Mini Kit (Qiagen), or TRIzol reagent (ThermoFisher Scientific). RNA isolated for downstream qRT-PCR or sequencing experiments was subjected to DNase treatment either on-column (Quiagen RNase Free DNase Set) or column-free (ThermoFisher Scientific TURBO DNA free kit). RNA was reverse-transcribed with the High-Capacity cDNA Reverse Transcription Kit (Applied Biosystems). qRT-PCR was performed using TaqMan probes and TaqMan Universal Master Mix II (ThermoFisher Scientific) on a QuantStudio 6 System. B-actin was used as a loading control for normalisation of target genes.

### Bisulfite pyrosequencing
One µg of total RNA per sample was bisulfite converted with the EZ RNA methylation kit, following manufacturer's instructions (Zymo Research). Reverse transcription and PCR amplification were performed with a PyroMark RT kit (QIAGEN) using primers optimised for the bisulfite modifications with one primer biotinylated. cDNA amplicons were enriched with streptavidin-coupled sepharose beads before following the manufacturer's recommendations for pyrosequencing.

### Northern blot analysis
An appropriate amount (0.5–2 µg) of total RNA was separated in 1% MOPS-formaldehyde agarose gels and transferred to Hybond-N + membranes (GE Healthcare). Membranes were exposed to either randomly [32P]-labelled dsDNA probes, [32P]-labelled strand-specific RNA probes, or with [32P]-end-labelled oligonucleotide probes, using RapidHyb (Sigma Aldrich). Membranes were exposed to a PhosphorImager screen, and the signal was quantified using a Typhoon FLA7000 system and ImageQuant TL 8.1 software (GE Healthcare). Primers used to generate dsDNA and oligonucleotide probes are listed in Supplementary Data 11.

### Oxford nanopore technology (ONT sequencing)
**Sample preparation for Nanopore direct long-read RNA sequencing.** RNA was isolated from isolated mitochondria from MEFs or skeletal muscle using the RNeasy Mini Kit (Qiagen) or RNeasy Fibrous Tissue Mini Kit (Qiagen) kits, following the manufacturer's instructions. Input quality control (QC) of samples was performed on an Agilent 2100 Bioanalyzer, using the Eukaryote Total RNA Nano kit to evaluate RIN values and concentration (Agilent). Dynabead polyA enrichment was performed on 6–18 µg of whole RNA, using the Invitrogen Dynabeads™ mRNA Purification Kit (ThermoFisherScientific Cat No 61006). RNA libraries were prepared as described in the Oxford Nanopore 'Direct RNA sequencing (SQK-RNA002)' protocol, version DRS_9080_v2_revU_14Aug2019, using the Oxford Nanopore Direct RNA Sequencing Kit (SQK-RNA002). 125–439 ng of polyA enriched RNA was used for reverse transcription, followed by ligation of

sequencing adapters. QC of the libraries was performed with the Qubit dsDNA HS kit (ThermoFisher Scientific). Finally, the samples were loaded on SpotON flow cells (FLO-MIN106), using the Flow Cell Priming Kit (EXP-FLP002), and sequenced on the Oxford Nanopore MinION system.

**Nanopore data analysis.** Fast5 files were merged using the multi_to_single_fast5 function from the ont_fast5_api toolkit (https://github.com/nanoporetech/ont_fast5_api/tree/master). The merged files were then re-basecalled using Guppy v4.4.1 with the following parameters: --flowcell FLO-MIN106, --kit SQK-RNA002, --recursive, --fast5_out, and --qscore_filtering 7. The resulting FASTQ files were mapped to the mitochondrial genome in the GRCm38 reference using Minimap2 v2.17 with parameters: -ax splice -uf -k14[75]. The SAM files generated from this mapping were converted to BAM format, then sorted and indexed using Samtools v1.10[76]. Next, the BAM files were converted to BED format using the bamtobed function from Bedtools v2.29.2[77]. The resulting BED files with same age and genotype were merged, processed, and visualised in R v4.3.1 using the ggplot2 package v3.4.3 for visualisation. The direct long-read Oxford Nanopore RNA sequencing raw data have been uploaded to NCBI SRA database with the BioProject accession number: PRJNA1192297.

**Calculation of unprocessed reads.** Reads were classified as unprocessed if their ends did not fall entirely within ±20 nt of a gene's annotated start or end site. Unprocessed reads were then grouped based on the gene boundaries they intersected. Specifically, reads spanning multiple gene boundaries were assigned to junction groups corresponding to all possible gene combinations (e.g. a read passing two genes was categorised under the junction group 'gene1,gene2'). To compare the proportion of unprocessed reads between knockout (KO) and control (Ctrl) conditions, we calculated:

$$\Delta Unprocessed\ reads_{junction1} = \\ 100 \left( \frac{unprocessed\ reads\ junction1_{ko}}{total\ reads_{ko}} - \frac{unprocessed\ reads\ junction1_{Ctrl}}{total\ reads_{Ctrl}} \right)$$
(1)

To evaluate the distribution of unprocessed junctions that were more prevalent in the KO condition, we first computed the percentage difference between KO and Ctrl for each junction. The positive values were then summed to obtain the total unprocessed signal. Each individual value was then expressed as a percentage of this total, and only junctions contributing at least 2% were included in the final visualisation. A pie chart was generated for each timepoint.

**Stable isotope labelling with amino acids in cell culture (SILAC) Stable isotope labelling.** KO and Control MEFs were grown for 19 days in DMEM for SILAC (ThermoFisher Scientific) supplemented with 10% Dialysed FBS, 200 mg/mL Proline (Sigma Aldrich), 50 μg/mL Uridine (Sigma Aldrich) and either heavy ($^{13}C_6$, $^{15}N_4$) or light L-Arginine and heavy ($^{13}C_6$, $^{15}N_2$) or light L-Lysine, followed by mitochondrial enrichment.

**Sucrose gradient fraction proteomics.** Mitochondrial pellets from labelled KO and Control MEF cells were enriched in MIB-C and quantified. 1.5 mg of heavy Arg/Lys KO and light Arg/Lys Control or heavy Arg/Lys Control and light Arg/Lys KO were combined, to give two replicate experiments each with a total of 3 mg of mitochondrial content. Merged mitochondria were lysed in MLB freshly supplemented with RNase Block (Agilent), as described for Western blot analysis The mixed mitochondrial lysate was loaded onto a linear 10–30% sucrose gradient and subjected to ultracentrifugation sedimentation. Fractions of 450 μL were taken and ¼ of fractions 1–19 were TCA precipitated for SDS-PAGE and Western blotting to establish the

mitoribosome sedimentation pattern. The remaining volume of fractions 9–11, 12–14 and 15–17, corresponding to mtSSU, mtLSU and monosome, was pooled and precipitated with three volumes of Ethanol. Protein pellets were resuspended in 20 μL of 6 M guanidine hydrochloride (GdmCL), 10 mM Tris-(2-carboxyethyl)phosphine hydrochloride (TCEP), 40 mM 2-chloroaceteamide (CAA), 100 mM Tris–HCl, pH 8.5. Overnight digestion and peptide cleaning were performed as described previously[78]. One third of the sample was used for LC-MS/MS analysis.

**LC-MS/MS analysis.** Peptides were separated on a 40 cm, 75 μm internal diameter packed emitter column (Coann emitter from MS Wil, Poroshell EC C18 2.7 micron medium from Agilent) using an EASY-nLC 1200 (ThermoFisher Scientific). The column was maintained at 50 °C. Buffer A and B were 0.1% formic acid in water and 0.1% formic acid in 80% acetonitrile, respectively. Peptides were separated at a flow rate of 300 nl / min, on a gradient from 6% to 31% buffer B for 57 min, from 31% to 44% buffer B for 5 min, followed by a higher organic wash. Eluting peptides were analysed on a Orbitrap Fusion Tribrid mass spectrometer (ThermoFisher Scientific). Peptide precursor m/z measurements were carried out at 60000 resolution in the 350 to 1500 $m/z$ range. The most intense precursors with charge state from 2 to 7 only were selected for HCD fragmentation using an isolation window of 1.6 and 27% normalised collision energy. The cycle time was set to 1 sec. The m/z values of the peptide fragments were measured at a resolution of 30,000 using an AGC target of 2e5 and 54 ms maximum injection time. Upon fragmentation, precursors were put on a dynamic exclusion list for 45 s.

**Protein identification and quantification.** Raw data were analysed with MaxQuant version 1.6.1.0[79]. Peptide fragmentation spectra were searched against the canonical and sequences of the *Mus musculus* reference proteome (proteome ID UP000000589, downloaded December 2018 from UniProt). Methionine oxidation and protein N-terminal acetylation were set as variable modifications; cysteine carbamidomethylation was set as fixed modification. Multiplicity was set to two; Arg10 and Lys8 were set as heavy labels. The digestion parameters were set to 'specific' and 'Trypsin/P', The minimum number of peptides and razor peptides for protein identification was 1; the minimum number of unique peptides was 0. Protein identification was performed at a peptide spectrum match and protein false discovery rate of 0.01. The 'second peptide' option was on. Differential abundance analysis was performed using limma, version 3.34.9[80] in R, version 3.4.3[81]. Protein groups annotated as Potential contaminant, Reverse, or Only identified by site were removed prior to the statistical analysis. The mass spectrometry proteomics data have been deposited to the ProteomeXchange Consortium via the PRIDE partner repository[82] with the dataset identifier PXD055907.

**Statistical analysis.** Limma's moderated two-sided t-test was used for significance testing based on two biological replicates, including a SILAC label swap. Benjamini and Hochberg's method to control the false discovery rate due to multiple testing was applied separately for each comparison to calculate the adjusted p-values. Proteins with adjusted p-values of less than 0.05 were designated as significant.

## Proteomics analysis
**Protein extraction and In-solution digestion.** Samples were thawed on ice, supplemented with 300 μL of 8 M urea (Sigma-Aldrich) in 50 mM ammonium bicarbonate buffer and homogenised with 50 mg of 400 μm silica beads using Vortex Disruptor Genie at 2800 rpm for 5 min. Then 150 μL of 0.1% ProteaseMAX (Promega) in 10% acetonitrile (ACN) was added and homogenisation was continued for 5 min. The samples were sonicated in a water bath for 5 min. Lysates were spun down at 14,000xg at 4 °C for 10 min and protein concentration was

determined by BCA assay (Pierce). An aliquot of 25 µg of each sample was reduced by adding 0.5 µL of 0.5 M dithiothreitol (DTT) at 25 °C for 1 h, alkylated with 1.5 µL of 0.5 M iodoacetamide for 1 h at room temperature (RT) in the dark. Tryptic digestion was started with 2 µL of 0.5 µg/µL lysyl endopeptidase (mass spectrometry grade, Waco, Japan) in an enzyme to protein ratio of 1:25 and incubated in a thermoshaker at 37 °C at 450 rpm for 3 h. Digestion was continued by adding 1 µg of sequencing-grade trypsin (Promega) and incubating overnight at 37 °C. The proteolysis was stopped with 2.5 µL concentrated formic acid (FA). The samples were cleaned on a C18 Hypersep plate (Thermo Scientific) and dried using a Vacufuge vacuum concentrator (Eppendorf).

**TMTpro labelling.** Dry samples were dissolved in 70 µL of 50 mM triethylammonium-bicarbonate (TEAB) and 100 µg of TMTpro reagents (Thermo Scientific) in 30 µL of anhydrous ACN were added. Samples were scrambled and incubated at RT at 450 rpm for 2 h. The labelling reaction was stopped by adding 11 µL of hydroxylamine at final concentration of 0.5% and incubated for at RT with 550 rpm for 15 min. Individual samples were combined to one analytical sample and dried in vacuum, followed by clean up on C18 StageTips and dried again.

**High pH reversed phase LC fractionation and RPLC-MS/MS analysis.** The TMTpro-labelled tryptic peptides were dissolved in 45 µL of 20 mM ammonium hydroxide and were loaded onto an Acquity bridged ethyl hybrid C18 UPLC column (2.1 mm inner diameter × 150 mm, 1.7 µm particle size, Waters), and profiled with a linear gradient of 5–60% 20 mM ammonium hydroxide in ACN (pH 10.0) over 48 min, at a flow rate of 200 µL/min. The chromatographic performance was monitored by sampling the eluate with a UV detector (Ultimate 3000 UPLC, Thermo Scientific) scanning at 214 nm. Fractions were collected at 30 s intervals into a 96-well plate and combined in 12 samples, concatenating 8-8 fractions representing peak peptide elution. Approximately 2 µg samples were injected in an Ultimate 3000 nano LC on-line coupled to a Q Exactive HFX hybride quadrupole-Orbitrap mass spectrometer (Thermo Scientific). The chromatographic separation of the peptides was achieved using a 50 cm long C18 Easy spray column (Thermo Scientific) at 55 °C, with the following gradient: 4–26% of solvent B (98% ACN, 0.1% FA) in 90 min, 26–95% in 5 min, and 95% of solvent B for 5 min at a flow rate of 300 nL/min. The MS acquisition method was comprised of one survey full mass spectrum ranging from $m/z$ 375 to 1500, acquired with a resolution of $R = 120,000$ (at $m/z$ 200) targeting $5 \times 10^6$ ions, followed by data-dependent HCD fragmentations of maximum 18 most intense precursor ions with a charge state 2+ to 7+, using 45 s dynamic exclusion. The tandem mass spectra were acquired with a resolution of $R = 60,000$, targeting $2 \times 10^5$ ions, setting isolation width to $m/z$ 1.4 and normalised collision energy to 33%, setting first mass at $m/z$ 110.

**Data analysis.** The raw files were imported to Proteome Discoverer v2.4 (Thermo Scientific) and analysed using the SwissProt mouse protein database with Mascot v2.5.1 (MatrixScience Ltd, UK) search engine. Parameters were chosen as follows: up to two missed cleavage sites for trypsin, 10 ppm peptide mass tolerance, and 0.05 Da for the HCD fragment ions. Carbamidomethylation of cysteine was specified as a fixed modification, whereas oxidation of methionine, deamidation of asparagine and glutamine, and TMTpro (+304.207) of lysine and N-termini were defined as variable modifications. For quantification, both unique and razor peptides were requested using reporter ion intensities. The mass spectrometry proteomics data have been deposited to the ProteomeXchange Consortium via the PRIDE partner repository[82] with the dataset identifier PXD062614.

**Statistical analysis.** Proteomic data were analysed using the limma package (v.3.38.3) in $R$ (v3.5.2), applying two-sided moderated $t$-tests. Multiple testing corrected $P$ values were calculated as FDR, with FDR < 0.05 defining significance.

## Cryogenic electron microscopy (cryo-EM)
**Cryo-EM sample preparation.** Mitochondrial pellets from MEF cells were freshly enriched and then further clarified using a step gradient of 1 M and 1.5 M Sucrose in a buffer of 20 mM Tris and 1 mM EDTA. The bilayer step gradients were prepared in 14 × 89 mm Ultra-Clear Centrifuge Tubes (Beckman Coulter). Mitochondria were resuspended in MIB-C buffer and loaded onto the top of the step gradient, then spun in an Optima L-80 XP Ultracentrifuge (Beckman Coulter SW41-Ti rotor). at 25,000 rpm for 1 h at 4 °C. Clarified mitochondria sedimented as a band at the interface of 1 M and 1.5 M sucrose, which was then collected and diluted with an equal volume of 10 mM Tris-HCl pH 7.4. The samples were subjected to a final 10 min cold spin at 11,000 × $g$ and then resuspended in a freezing buffer containing 300 mM Trehalose, 10 mM Tris-HCl pH 7.4, 10 mM KCl, 1 mM EDTA, 0.1% BSA for long-term storage at −80 °C.

Mitochondria were thawed and resuspended in 2% b-DDM ($n$-dodecyl β-d-maltoside), 25 mM HEPES, 10 mM Mg(OAc)$_2$, 50 mM KCl, 1 mM DTT, and protease inhibitor cocktail (Roche), incubated on ice. The mitochondrial suspension was then subjected to homogenisation with 10-20 manually administered strokes in a glass homogeniser. The solution was kept on a rocker at 4 °C for 20 min. The lysate was layered on top of 0.8 M sucrose solution buffered with 25 mM HEPES, 10 mM Mg(OAc)$_2$, 1 mM DTT, and 1% b-DDM in a TLA120.2 thick-walled tube and subjected to ultracentrifugation at 100,000 rpm in a TLA 120.2 rotor (Beckmann Coulter) for 1 h. The resulting pellet was then gently washed and resuspended in 25 mM HEPES, 10 mM Mg(OAc)$_2$, 50 mM KCl, 1 mM DTT, and 0.02% b-DDM. The solution was clarified twice by centrifugation at 15,000 × $g$ for 15 min, and the final solution was used for cryo-EM grid preparation.

To capture more assembly states, including those with bound GTPases, mitochondria were lysed using the same protocol as above, with the addition of RNase inhibitor and 0.5 mM GMPPNP (a non-hydrolysable GTP analog) to the lysis and resuspension buffers. Further, to prevent potential degradation of sensitive complexes due to blotting and vitrification, 0.5 mM BS3 (Thermofischer A39266) was added 10 min prior to grid freezing. BS3 is a crosslinking agent (11 Å spacer) that can form irreversible bonds between a pair of amino-groups of lysine-side chains and/or N-terminal residues. Comparable concentrations of BS3 as here have been used in structural studies of the mitoribosome[83] and spliceosome[84]. The sample was then clarified with centrifugation at 15,000 × $g$ for 2 min before being used. A volume of 3 µL sample (A$_{260}$ 1.64) was placed on a glow discharged (for 30 s at 20 mA) Quantifoil Au 300 mesh R2/1 grid coated with ~3 nm thick continuous carbon support in a controlled environment of 100% humidity and 4 °C temperature using Vitrobot mKIV (FEI/Thermo Fisher). After 30 s of incubation, the excess sample was blotted off for 3 s and subjected to vitrification in liquid ethane. The frozen grids were then stored in liquid nitrogen for data collection.

**Cryo-EM data acquisition and image processing.** Data acquisition was carried out using a FEI Titan Krios (FEI/ThermoFisher Scientific) 300 kV electron microscope equipped with a Gatan K3 detector. For the untreated sample, a total of 65,125 movies (dataset 1) were collected (35 frames each) at a defocus range of −0.5 to −1.8, an electron dose rate of 1.05 e/frame/Å$^2$ at a pixel size of 0.846 Å (105,000 × magnification). For the treated sample, two cryo-EM datasets (2 and 3) were acquired. Dataset 2 had a total of 47,960 movies (35 frames each) collected at a defocus range of −0.5 to −1.8, electron dose rate of 1.05 e/frame/Å$^2$, with pixel size of 0.846 Å$^2$. Dataset 3 had a total of 60,000

movies (35 frames each) collected at a defocus range of −0.5 to −1.8, electron dose rate of 1 e/frame/ $Å^2$, with pixel size of 0.825 Å.

Motion correction was carried out in Relion 3.1.1[85] using its implementation of MotionCorr 2.0[86]. The motion-corrected micrographs were used for CTF estimation with GCTF[87]. Particles were exhaustively picked by blob picking and separately with mtSSU, mtLSU, and monosome class averages used as references. The picked particles were extracted at a box size of 540 pixels binned to 180 pixels, and subjected to 2D classification. We did not obtain mtSSU as the purification protocol was optimised for mtLSU/monosome enrichment using a 0.8 M sucrose cushion buffer. The particle images from classes corresponding to the large subunit/assembly states were pooled to produce a consensus map with 3D-autorefinement. The data was further cleaned up with 3D classification with local angular search range of ±1.8°, and the junk images were rejected. These were then subjected to per-particle CTF correction (beam-tilt, per-particle defocus, per-micrograph astigmatism) followed by Bayesian Polishing and a second round of CTF refinement (beam-tilt, per-particle defocus, per-micrograph astigmatism, trefoil and fourth-order aberrations, magnification anisotropy). At this stage, the particles from datasets 2 and 3 were pooled together. The particle images were now binned again 180 pixels, and subjected to a series of unaligned focused 3D classifications with signal subtraction (FCwSS) on the interface and around factor binding regions to isolate distinct mtLSU assembly intermediates. Particles from dataset 1 (native) classified into MRM3-dimer states (unsorted states A1 to A4), NSUN4-MTERF4 bound states with/without MRM2 and GTPBP7 in open conformation (states C2, D1 + D2). Particles from datasets 2 and 3 were classified into two states with GTPBP7 (closed conformation) (state C1) and GTPBP7 (closed conformation) and GTPBP10 (unsorted states B1 and B2). The particles corresponding to identified assembly intermediates were then unbinned to 540 pixel box size and used for 3D auto-refinement to produce the final assembly state maps (Supplementary Fig. 12). MRM3-dimer bound states were further sorted into four subsets based on the occupancy of DDX28, followed by uL16m by performing FCwSS on the central protuberance and uL16m, respectively (sorted states A1, A2, A3 and A4). Similarly, unaligned FCwSS was done by masking uL16m to sort GTPBP10-bound states based on the occupancy of uL16m (sorted states B1 and B2). MRM2 bound state (D1 + D2) was classified further into GTPBP7-bound (state D1) and unbound states (state D2) by performing masked FCwSS on GTPBP7 binding region. The maps were sharpened with an appropriate *B*-factor and local-resolution filtered (Supplementary Fig. 12). The maps with MRM3 dimer were at relatively lower resolution and suffered from substantial anisotropy due to preferred orientation. The corresponding particle images were exported to CryoSparc v4.1[88] and subjected to nonuniform refinement, which significantly reduced map anisotropy. The maps thus obtained for states A1-A4 had an overall resolution of 5.67 Å, 4.363 Å, 7.31 Å and 7.42 Å, respectively. The overall resolution for states B1 and B2 was 3.62 and 3.08 Å, respectively; for states C1 and C2, it was 3.11 Å and 2.95 Å, respectively; finally, for state D (D1 + D2), it was 2.91 Å. For generating a good quality map for CP-tRNA, particle images from NSUN4-MTERF4-bound states (in which the CP has matured) were pooled and subjected to 3D autorefinement, yielding a map of ~3.1 Å overall resolution (Supplementary Figs. 11–13). The map was then *B*-factor sharpened and local-resolution filtered to be used for the identification and modelling of the CP-tRNA species. The data acquisition parameters and data processing details are listed in Supplementary Data 12.

**Model building and refinement**. Manual model building was done using PDB 7O9M[89] as the starting template for NSUN4-MTERF4 bound states and PDB 7OI6[90] as the template for MRM3 dimer bound states. First, the models were rigid-body fitted into the respective maps in UCSF Chimera v1.14[91]. Next, each chain was individually rigid-body

fitted into the individual maps in Coot v0.9.1[92]. The sequences for mouse *16S* rRNA and mitoribosomal proteins were taken from NC_005089. The sequences of the rRNA and protein chains in the structures were mutated in Coot v0.9.1 using the inbuilt 'Align and mutate' function. The alignment and mutations generated were manually checked against the map and corrected wherever required.

CP-tRNA model was built by docking the model of aminoacylated *mtV* (from PDB 7QI4[93] into the local resolution filtered map of the central protuberance. The sequence was then mutated to mouse *mtV* and real-space refined into the map with significant remodelling of the elbow region. The density revealed an incomplete match between the bases and the density. Mutating the sequence to *mtF* provided a much better agreement of the bases with the density (Fig. 5b). However, there is an extra density at the acceptor arm of the tRNA, which can accommodate a single nucleotide. Since *mtV* is one residue longer than *mtF*, it potentially indicates the presence of *mtV* at the CP, however, at a lower occupancy as compared to *mtF*. Therefore, a molecule of *mtF* was modelled here and retained in all assembly states in this work. We observed no density for 3′ CCA (Fig. 5b).

For MRM3-bound states (A1-A4), models were built for A1, A2 and A3, while A4 was skipped due to poor overall resolution and being generally similar to A3 except for the lack of uL16m. The overall resolution for these states (A1-A3) allowed us to check for occupancy and confirmation of protein and rRNA chains, but not check residue side chain rotamers, ligands, and metal ions. The chains for DDX28, uL16m, and bL36m were removed in the states that lacked the corresponding density. None of these states had a density for GTPBP10, contrary to the previous observation. Hence, the chain was removed and H89 was remodelled depending on the presence or absence of uL16m. Some densities were left unmodelled as the poor resolution did not allow identification. The chains were then subjected to self-restrained real-space refinement individually. For NSUN4-MTERF4-bound states (B1, B2, C2 and D), the models were subjected to real-space refinement, and agreement with the map was checked and ensured in a residue-by-residue manner for most of the chains with sufficient local resolution. Specifically, for states B1 and B2, the initial model for GTPBP10 was taken from AlphaFold Protein Structure Database[94,95] (AF- Q8K013-F1) and fitted into the map. Due to relatively poor local resolution, the model was fitted with coot generated reference-restrained real-space refinement to ensure a good agreement with the density on the level of secondary structure. The model for GTPBP7 was taken from PDB 7PD3[60]. uL16 m was removed and H89 was remodelled to agree with the density maps of states B1 and B2. A molecule of GMPPNP was placed in the binding pockets of GTPBP7 and GTPBP10. For state C1, GTPBP10 was removed. For state D1, particles from states D1 and D2 were pooled to improve the quality of the map; GTPBP7 was partly retained in the open conformation as in PDB 7O9M. A molecule of SAH was placed in the binding pocket of MRM2, as a distinct density could be observed. Real-space refinement of the models was carried out with torsion and Ramachandran restraints. Restraints for ligands and rRNA modifications were generated from the CCP4 7.0 library[96] and imported into coot for model building and refinement. Models were not built for states A4 and C2 due to low resolution or high anisotropy.

The models were then hydrogenated using ReadySet from the PHENIX suite[97]. The output models were further refined against the respective *B*-factor sharpened, local-resolution filtered maps by carrying out energy minimisation and ADP (atomic displacement parameter) estimation with rotamer and Ramachandran restraints using Phenix.real_space_refine v1.21[97] followed by validation with MolProbity[98].

## Reporting summary
Further information on research design is available in the Nature Portfolio Reporting Summary linked to this article.

## Data availability

Correspondence and material requests should be addressed to the corresponding authors. All biochemical data supporting this study are available within the paper, its supplementary information and its source data files. All mass spectrometry proteomics datasets are deposited to the ProteomeXchange Consortium via the PRIDE partner repository with the dataset identifiers PXD055907 (SILAC gradient proteomics), PXD062614 (Skeletal muscle proteomics) and PXD019550 [http://proteomecentral.proteomexchange.org/cgi/GetDataset?ID=019550] (MEF proteomics). Direct long-read Oxford Nanopore RNA sequencing raw data have been deposited to NCBI Sequence Read Archive (SRA) with the BioProject accession number: PRJNA1192297. Atomic models and cryo-EM maps have been deposited to Protein Data Bank (PDB) and Electron Microscopy Data Bank (EMDB) with the following IDs: 9HCC and EMD-52044 (State A1); 9HCD and EMD-52045 (state A2); 9HCE and EMD-52046) (state A3); EMD-52408 (state A4); 9E9C and EMD-47791 (state B1); 9HCF and EMD-52047 (state B2); EMD-52430 (state C1); 9HCH and EMD-52049 (state C2) and 9HCG and EMD-52048 (state D). Map quality of states A4 and C1 was inadequate due to low resolution or high anisotropy and models were not built. Source data are provided with this paper.

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

## Acknowledgements

This study was supported by grants to A.Wr. from the Swedish Research Council (VR2022-01287 and VR2023-07091), the NovoNordisk Foundation (NN0082202), the Knut and Alice Wallenberg Foundation (KAW2019.0109), the Region Stockholm (RS2022-0708), a Karolinska Institutet consolidator grant (2-190/2022), Heart and Lung Foundation (20210498) and Cancerfonden (21 1621 Pj). A.A. was supported by the European Research Council (ERC-2018-StG-805230). Protein identification and quantification were carried out by the Proteomics Biomedicum Core Facility, Karolinska Institutet https://ki.se/en/mbb/proteomics-biomedicum. The authors would like to acknowledge support of the National Genomics Infrastructure (NGI) / Uppsala Genome Center and UPPMAX for providing assistance in massive parallel sequencing and computational infrastructure. Work performed at NGI / Uppsala Genome Center has been funded by RFI / VR and Science for Life Laboratory, Sweden. Cryo-EM data were collected at the Swedish National Cryo-EM Facility, SciLifeLab, Stockholm University, and at the Karolinska Institutet 3D-EM Core facility. The inhibitor of mitochondrial transcription (IMT1B) was a kind gift from N-G Larsson, Karolinska Institutet.

## Author contributions

Conceptualisation, R.I.C.G., C.F. and A.Wr; investigation R.I.C.G.; V.S., L.P-P., A.Wi., M.F.M., D.M., F.A.R., X.L., M.S., M.C., A.A., C.F. and A.Wr. formal data analysis R.I.C.G., V.S., L.P-P., M.F.M., I.A., C.F., A.Wr. funding acquisition, A.We, J.R., A.A. and A.Wr.; writing original draft, R.I.C.G., V.S., C.F. and A.Wr.; reviewing, all authors.

## Funding

## Competing interests

The authors declare no competing interests.
