## [Transparent Peer Review file · Nature Communications]

The mitochondrial methylation potential gates mitoribosome assembly

Corresponding Author: Professor Anna Wredenberg

Version 0:

Reviewer comments:

Reviewer #1

(Remarks to the Author)

Summary

Glasgow et al. use a combination of direct RNA sequencing, proteomics, and structural analysis by Cryo-EM to determine the role of RNA methylation on mitochondrial gene expression by knocking out the mitochondrial SAM importer in both skeletal muscle in mice and in MEFs. While this is a bit of a sledgehammer approach, the authors demonstrate that it is an effective one, as the number of methyltransferases involved in these pathways is extensive. The authors observe processing defects in the polycistronic RNA, despite previous studies showing that methylation is not required for RNA cleavage, although the mechanism for this remains unclear. Loss of methylation differentially affected specific tRNAs, with some becoming undetectable and others increasing. Processing of the rRNA cluster is particularly affected. The authors use a combination of gradient analysis and protein profiling with structural studies to show that ribosome assembly is stalled early on, and that methylation is needed for maturation of the peptidyl transferase center. The data presented in this study are both compelling and extensive, coming from both conditional KO mice and from cell culture and assessing numerous abundances and properties of mitochondrial RNA and proteins. The text and figures are generally clear, although there are a few changes that could be made (both in the text and aesthetically) that could improve clarity. I also think that additional analysis of RNA cleavage site defects (or just additional plotting of performed analyses) could be beneficial. Overall, the manuscript highlights the importance and role of methylation in multiple gene expression pathways in mitochondria (primarily RNA processing and ribosome assembly). Future studies can build upon this to determine the mechanism and the importance of particular MTases/methylation sites on these processing steps.

Major comments:

(1) The authors use both a conditional skeletal muscle KO and a tissue culture MEF KO of Slc25A26, the mitochondrial SAM transporter, to understand the role of methylation in mitochondrial gene expression regulation. The observed phenomenon are generally consistent between the two systems, but there are some interesting observations that are never discussed. Are the differences between the two thought to primarily be due to heterogeneity of cell types within the harvested quadriceps tissue (i.e. more than just skeletal muscle)? Or does this have to do with when in development Mlc1f-Cre turns on in these cells? Or a different explanation? It would be helpful to comment on this somewhere.

(2) The authors observe a processing defect, particularly in the rRNA region, but also around ND2 and ND4/4L. The difference in processing can be observed in both the nanopore data in 1e and 2a. Given the complexity of the system (e.g. TRMT10C methylates some tRNAs but not all, some processing sites are flanked by tRNAs but not all, etc), it might be interesting to quantify and plot from the ONT the processing defects at each site. For example, the change in the fraction of reads unprocessed at each site between control and KO tissue/cells. This might make it visually clearer which sites, and to what extent, are affected by a loss of methylation.

Minor comments:

(1) PTC is never defined in the manuscript as "peptidyl transferase center". This should be done somewhere (e.g. line 83).

(2) Line 95: This should be "bisulfite sequencing" instead of "bisulphate"

(3) Figure 1b: Why are ATP6, ATP8, ND1, ND2, ND5, ND6, and CYTB not present in the MS data?

(4) Figure 1c: I would write COX1, COX2, COX3 on the X-axis instead of CI, CII, and CIII to avoid confusion with Complex I, Complex II, and Complex III.

(5) Figure 1d: The font for the individual tRNAs are hard to read. Could you just write the amino acid (e.g. Ala) and above write "tRNA"?

(6) Line 136: "accumulated to a large proportion of the total reads" – can you indicate the proportion here in parentheses?

(7) Figure 2a: The yellow chosen here for close to 0 is very hard to see, in particular when printed. Also, I'm not sure I understand what the legend means by "of first transcript".

(8) Figure 2b-d: The "Fraction of reads" line thickness bar doesn't provide much information here since the lines here are roughly the same size. I believe this is used here to match Supplementary Figures 3, 4, and 5. That plotting works better in the Supplement where the figures are larger. For the main figure, is it possible to just do fraction of reads? You could also limit the color scale here to not go all the way from 0 – 1 if you're trying to make a point about the fraction of reads that match each of the illustrated processed forms.

(9) Figure 3d: It's hard to visually process all of the changes here because of the number of samples and proteins being blotted. Can you either (a) separate 8wk, 12wk, and MEF or (b) quantify the bands and plot as a bar graph?

Reviewer #2

(Remarks to the Author)

The manuscript by Glasgow et al. provides a very elegant work on the role of methylation for mitochondrial gene expression. The authors established a conditional skeletal muscle-specific mouse model and mouse embryonic fibroblasts (MEFs) deficient in mitochondrial methylation by deleting the mitochondrial SAM carrier (SAMC). These tools allow the authors to investigate the vital role of methylation *in vivo* and *in vitro* specifically within mitochondria without affecting the responsible methyltransferases. The authors convincingly show that mitochondrial RNA precursors are not efficiently processed without SAM especially the gene cluster containing tRNA^{Val}-12S-tRNA^{Phe}-16S leading to an accumulation of rRNA precursors. Interestingly, those rRNA precursors seem to be associated with processing and ribosome biogenesis factors as indicated by sucrose gradient analysis. Using a combination of biochemical approaches, mass spectrometry and cryo-electron microscopy Glasgow et al. reveal the consequences of mitochondrial SAM depletion on mitoribosome assembly. Ten immature mtLSU complexes were solved with different assembly factors bound. This set of biogenesis intermediates reveal alternative assembly pathways. Interestingly, murine mtLSUs contain equally mt-tRNA^{Val} and mt-tRNA^{Phe}, however, the immature mtLSU complexes accumulating in the knockout reveal a shift towards mt-tRNA^{Phe} lacking CCA addition at the 3' end indicating that this modification is not required for the incorporation of the structural tRNA at the CP. The manuscript is very well written and the figures are of high quality. The study will be of high interest for a broad readership and is suitable for publication in Nature Communications. I have only a few points, which the authors may want to address:

1. Supplementary Fig. 9a / page 6, line 172-174

The authors claim that the sedimentation pattern is normal and that the modified tRNA composition is the reason for the translation defect in Skm-KO. However, in the opinion of the reviewer there are differences between Ctrl and KO. The Ctrl sample shows defined peaks for SSU, LSU and monosome, whereas the signals for uS15m and mL45 tend to tail through the gradient from the SSU and LSU fractions without the defined peak in the monosome fraction. This indicates a problem in monosome formation often accommodated with elevated signals of MRPs. This is not in conflict with the story, it rather supports the vital role of methylation for ribosome biogenesis *in vivo*.

2. Fig. 3b / page 6, line 174-175

The authors claim "normal signals corresponding to the mtLSU and mtSSU" in MEFs KO. However, there is a clear shift of the signals for mL37 and uS15m by one fraction, which already indicates a defect in subunit biogenesis, which is in agreement with the solved immature LSU complexes. Also here, the authors are encouraged to put some more weight on their biochemical data.

3. Suppl. Fig. 9g

Change "Samc MEF Ctrl" to "Ctrl MEF" and "Samc MEF KO" to "KO MEF" to be consistent throughout the manuscript. The results presented in Suppl. Fig. 9g looks different to the results in Fig. 3b. Independent of the different gradient conditions, in Suppl. Fig. 9g monosomes seem to be present in contrast to Fig. 3b. The authors may want to revisit the figure.

minor points

4. page 7, line 213

"...LSU and the monosome (fractions 12-15) (Fig. 4d)."

I think this should be changed as followed: "...LSU (fractions 10-12) and the monosome (fractions 13-15)"

5. page 8, line 232

change "16s" to "16S"

6. page 9, line 259

delete "ref"

7. page 10, line 287-288

remove links and delete "ref"

Reviewer #3

(Remarks to the Author)

The mitochondrial methylation potential gates mitoribosome assembly

by Glasgow et al.

The authors investigate the role of S-adenosyl methionine (SAM) in mitochondrial gene expression, focusing on its impact on mitochondrial ribosomal RNA (rRNA) processing and ribosome assembly. Using mouse embryonic fibroblast (MEF) cells depleted of the SAM carrier (SAMC) and quadriceps cells from mice with a skeletal muscle-specific deletion of *Slc25a26* (encoding SAMC), they employ long-read RNA sequencing, quantitative mass spectrometry (label-free and stable isotope labeling), sucrose gradient sedimentation, and cryo-EM to assess the consequences of SAM depletion.

Their findings demonstrate that mitoSAM deficiency disrupts mitochondrial ribosome assembly by impairing rRNA processing, stalling subunit maturation, affecting structural tRNA incorporation, and preventing proper peptidyl transferase center (PTC) formation. These results underscore the critical role of mitoSAM in mitochondrial gene expression and ribosome biogenesis.

While the study is methodologically sound and presents compelling data, the authors could better incorporate and discuss relevant previous research in this area.

Major Points:

There are several relevant references that I would like to bring to the authors' attention. For example, Ishiguro (2019, NAR) showed that SAM depletion in *E. coli* causes a growth defect that can be rescued by overexpression of the methyltransferase RlmE, the likely bacterial homolog of MRM2. Similarly, Arai et al. (DOI: 10.1073/pnas.1506749112) and Wang et al. (10.1073/pnas.1914323117) described structural and mechanistic consequences of RlmE depletion, resulting in LSU precursors with structural deficits comparable to those reported here. Additionally, Nikolay et al. (10.1016/j.molcel.2021.02.006) identified a natural LSU precursor containing ObgE - GTPBP10 is one of the two ObgE homologs - with incomplete maturation in similar regions of 23S rRNA (H90-93, H89, H80). Finally, Lai et al. (10.1371/journal.pone.0090818) identified Ftsj2 (MRM2, Uniprot ID Q9CPY0) as the mitochondrial homolog of RlmE. The authors may incorporate these references along with any other relevant studies and discuss how their findings align with or extend prior work. The discussion is currently brief and would benefit from a more thorough comparison with these studies.

Minor Points:

The phrasing in some instances should be clearer and more precise:

- Line 31: "...required during peptidyl transferase center formation and mitochondrial ribosome assembly." → PTC formation is part of ribosome assembly. This should be reworded for accuracy.
- Line 32: "Our data thus identify a critical role for methylation at two steps during mitochondrial gene expression." → What are these two steps? This should be explicitly stated.
- Line 74: "... and the deletion of SAMC in mice and flies is embryonic and larvae lethal. → The sentence is slightly awkward. Please rephrase e.g., "...and SAMC depletion in mice and flies results in embryonic and larval lethality.
- Line 93: "Born at Mendelian ratios and appeared normal at weaning but needed to be sacrificed by 12 weeks of age due to plateaued weight." → This should be reworded for clarity and provide more context for readers unfamiliar with mouse development.
- Line 96: "Quadriceps presented with less than 50% methylation." → Consider rephrasing as: "The quadriceps muscle exhibited a 12S rRNA methylation level of less than 50%."
- Line 100: Please define OXPHOS.
- Lines 121ff: What causes the accumulation of incomplete transcripts? Why do they pile up?
- Line 160: Please specify which mitochondrial transcription inhibitor was used.
- Line 164: "Thus, this data identify methylation within the rRNA gene cluster as critical for efficient processing." → Does this mean "critical for its efficient processing"? Please clarify.
- Line 171: "We performed sedimentation experiments on the mitochondrial ribosome." → This phrasing is awkward; please reword.
- Line 205: "Additionally, early processing and maturation factors in the monosome fractions indicate that a larger protein-RNA complex already assembles prior to complete cleavage of the ribosomal gene cluster." → This sentence is unclear; please rephrase.
- Line 221: "Thus, our results indicate that in the absence of mitoSAM, 12S and 16S rRNAs initiate early stages of assembly but stall due to a lack of processing." → A lack of processing of what? Please specify.
- Line 273: "GTPBP7 was bound in a fluid conformation..." → "Fluid conformation" is vague; consider using more precise terminology.
- Line 291: "Methylation is of particular importance at two sites during mitochondrial gene expression, with both rRNAs and tRNAs possessing several modifications." → What are these two sites? This should be specified.
- Line 310: "Our findings also show that the 2'-O-methylations of 16S rRNA are crucial for the maturation of the PTC and LSU. → Similar to Line 31. Please rephrase, as the PTC is part of the LSU.
- Figure Labels: In several figures (e.g., Fig. 2a, e; Fig. 4d; Fig. 5; Fig. 6b; Supp. Fig. 9e, f), the labeling text is too small. Ensure that the final scaled version maintains a minimum font size of 6 points.
- Fig. 6b: Verify and correct the color coding for the A loop and P loop.
- Fig. 9e, f: These Figs. are overcrowded and difficult to interpret. Consider an alternative visualization or highlighting only a subset of peptides/ proteins. Also, were these analyses repeated? If so, indicate sample size (n=x).

• Supplementary Figures: Many supplementary figures, particularly Supp. Figs. 3–8, have insufficient legends. More detailed figure legends should be provided.

Reviewer #4

(Remarks to the Author)
NCOMMS-24-74310-T

Summary:

The manuscript by Glasgow et al. proposes two roles for SAM in mitochondrial gene regulation. Direct long-read RNA sequencing highlights the requirement for SAM in efficient RNA processing (particularly the rRNA cluster). Profiling of ribosome fractions identifies the enrichment of RNA processing factors and mitoribosome assembly factors suggesting a defect in mitoribosome production. Cryo-EM studies reveal 10 distinct intermediate assemblies (immature LSU classes) where notably the peptidyl transferase center appears disordered. This manuscript was very well written. While the results presented in this work are important for the field, there is key information and context missing. This is particularly true when considering the relative low resolution of the cryoEM data and the associated assertions made on the molecular basis of LSU maturation.

Please find specific comments below.

1. The authors place a strong focus on tRNA containing processing sites (likely due to the known methylation of mito-tRNAs). However, there is a short statement in the discussion that there was no impact on atypical processing sites (sites without tRNAs). This selectivity appears to be an important point and the data should be incorporated into a Supplementary Figure.
2. The authors discuss the role of SAM in processing efficiency, but do not comment on observed changes in processing fidelity or alternative processing sites. The authors should comment on whether SAM exclusively impacts cleavage efficiency of canonical sites or whether other spurious RNA processing events may also influence translation dysregulation.
3. The authors state in the introduction that methylation of mito-tRNAs is not necessary for precursor RNA processing by RNase P and RNase Z. Therefore, the molecular mechanism remains unclear. Did the authors measure RNase P/Z abundance to see whether changes in SAM import impact the levels of these endoribonucleases?
4. The manuscript lacks important information about the quality of the structural data presented in this work. For instance, Supplementary Figures 10 and 11 demonstrate a clear distinction in the map quality and interpretability across the 10 reconstructions. The authors need to include a description of the overall quality of these maps in the results section prior to discussing their interpretations. In areas of particular focus (e.g. PTC, CP), the authors should describe the local resolution and what can and cannot be inferred from these regions of the map.
5. Supplementary Table 12 is missing from the submitted files. This is imperative for readers to discern the confidence one can place on the conclusions made from each map/model.
6. The authors state “Surprisingly, no 3’CCA RNA modification was present in all mature tRNAs, suggesting that this modification is not required for incorporation into the mitochondrial ribosome” (lines 239-241). This conclusion appears to be motivated by the lack of density at the 3’-end of the tRNA presented in Figure 5b. The authors need to specify which map is displayed and what is the local resolution in this area of the map. The resolution appears insufficient to make such a definitive claim.
7. Similar to point 6, the authors state in lines 231-233 that the cryo-EM maps lack features to describe methylation groups. While this may be true, the resolution in this region of the map appears to be insufficient to state definitively that there are no RNA modifications. The authors should be cautious in this interpretation.
8. The authors leverage chemical crosslinking to stabilize LSU intermediates. The authors should include this important point in the results and demonstrate how they optimized or verified the crosslinker is not influencing observed distortions.
9. Were the authors able to observe SSU intermediates? It would be informative to know whether SAM also impacts SSU maturation and how.

Minor points:

Please label Figure 2, panel E.

Supplementary table listing SamC mouse genotyping primers appears incomplete. Please revise.

Supplementary Figure 10 has replicated typos “re finement”

Version 1:

Reviewer comments:

Reviewer #1

(Remarks to the Author)

The authors have addressed my concerns. Their re-analysis of their ONT data clearly shows the effects on the individual processing sites and could be a useful pipeline for further studies dissecting the roles of specific modifications on mtRNA processing. They also expanded analysis and discussion of mitoribosome assembly defects and compared their results to phenomena observed in *E. coli*. I think the manuscript will be of great interest to the fields. I have only a few minor comments:

- The title and y-axis label in Figure 2C could be more streamlined. Potentially “Unprocessed read differences (KO-Ctrl)” for the title and “% difference” or “Percent difference” for the Y-axis.
- Supplementary Figure 5b: It would be nice to have a table title so it's understandable without the caption
- Line 298: “CP” introduced but not defined as “central protuberance”

Reviewer #2

(Remarks to the Author)

The authors have addressed all issues. In my opinion the revised manuscript is suitable for publication in Nature Communications.

Reviewer #3

(Remarks to the Author)

The authors have done a great job improving their manuscript, including addressing the comments and suggestions from the other reviewers, which— from my perspective—were handled very well.

While reviewing the revised version, I noticed a minor typo:

- Fig. 6b: acceter arm → acceptor arm

Overall, the scientific quality, data presentation, and readability have improved significantly. In my opinion, this is a very nice piece of work that deserves publication in Nature Communications.

Reviewer #4

(Remarks to the Author)

The authors have revised the manuscript by Glasgow et al. to address the concerns and suggestions raised by the reviewer. The major changes include a reanalysis of ONT sequencing data to provide additional insight into methylation-dependent RNA processing and an improved description of the quality and interpretability of the cryoEM reconstructions. The revised text and new figures present a compelling story that is important for the field. The authors addressed concerns related to the structural study, and I support its publication.

Response letter

REVIEWER COMMENTS

We would like to thank all reviewers for their helpful comments. We have made sincere efforts to address all concerns raised. We have re-analysed the ONT sequencing data and now present new figures (Fig.2, S3, and S4). We also extensively expanded the description of the structural data, as well as the discussion. Please find below our individual responses.

Reviewer #1 (Remarks to the Author):

Summary

Glasgow et al. use a combination of direct RNA sequencing, proteomics, and structural analysis by Cryo-EM to determine the role of RNA methylation on mitochondrial gene expression by knocking out the mitochondrial SAM importer in both skeletal muscle in mice and in MEFs. While this is a bit of a sledgehammer approach, the authors demonstrate that it is an effective one, as the number of methyltransferases involved in these pathways is extensive. The authors observe processing defects in the polycistronic RNA, despite previous studies showing that methylation is not required for RNA cleavage, although the mechanism for this remains unclear. Loss of methylation differentially affected specific tRNAs, with some becoming undetectable and others increasing. Processing of the rRNA cluster is particularly affected. The authors use a combination of gradient analysis and protein profiling with structural studies to show that ribosome assembly is stalled early on, and that methylation is needed for maturation of the peptidyl transferase center. The data presented in this study are both compelling and extensive, coming from both conditional KO mice and from cell culture and assessing numerous abundances and properties of mitochondrial RNA and proteins. The text and figures are generally clear, although there are a few changes that could be made (both in the text and aesthetically) that could improve clarity. I also think that additional analysis of RNA cleavage site defects (or just additional plotting of performed analyses) could be beneficial. Overall, the manuscript highlights the importance and role of methylation in multiple gene expression pathways in mitochondria (primarily RNA processing and ribosome assembly). Future studies can build upon this to determine the mechanism and the importance of particular MTases/methylation sites on these processing steps.

We thank the reviewer for their insight and genuinely useful suggestions to improve the manuscript.

Major comments:

(1) The authors use both a conditional skeletal muscle KO and a tissue culture MEF KO of Slc25A26, the mitochondrial SAM transporter, to understand the role of methylation in mitochondrial gene expression regulation. The observed phenomenon are generally consistent between the two systems, but there are some interesting observations that are never discussed. Are the differences between the two thought to primarily be due to heterogeneity of cell types within the harvested quadriceps tissue (i.e. more than just skeletal muscle)? Or does this have to do with when in development Mlc1f-Cre turns on in these cells? Or a different explanation? It would be helpful to comment on this somewhere.

This is an important point. The most plausible explanation for the differences between muscle tissue and MEF cells is the temporal and contextual aspect that is specific to the in vivo model. In the mouse model, we observe a progressive loss of mitochondrial SAM (mitoSAM), which triggers tissue-specific responses. It is reasonable to assume that the muscle will attempt to compensate for the declining mitoSAM levels and the associated metabolic and molecular changes. However, these responses eventually fail, and the animals cannot survive long enough to observe the consequences of a chronic mitoSAM depletion. In contrast, MEF cells represent a chronic state of mitoSAM depletion, allowing us to study a terminal state, not achievable in vivo. We have clarified this in the results section (line 118-122).

(2) The authors observe a processing defect, particularly in the rRNA region, but also around ND2 and ND4/4L. The difference in processing can be observed in both the nanopore data in 1e and 2a. Given the complexity of the system (e.g. TRMT10C methylates some tRNAs but not all, some processing sites are flanked by tRNAs but not all, etc), it might be interesting to quantify and plot from the ONT the processing defects at each site. For example, the change in the fraction of reads unprocessed at each site between control and KO tissue/cells. This might make it visually clearer which sites, and to what extent, are affected by a loss of methylation.

Thank you for this helpful comment. We have expanded on our representation of the processing sites with additional figures. We now present additional analysis of the ONT data, concerning processing frequencies for 5' and 3' gene junctions, as well as any additional processing sites. The new data is shown in figures 2, S3, and S4.

Minor comments:

(1) PTC is never defined in the manuscript as “peptidyl transferase center”. This should be done somewhere (e.g. line 83).

We have added the definition of PTC (line 96).

(2) Line 95: This should be “bisulfite sequencing” instead of “bisulphate”

We have corrected the spelling throughout the manuscript.

(3) Figure 1b: Why are ATP6, ATP8, ND1, ND2, ND5, ND6, and CYTB not present in the MS data?

Peptides for these mtDNA-encoded proteins were not identified in the sample preparation and MS approach we used. This is not unusual for these hydrophobic membrane subunits.

(4) Figure 1c: I would write COX1, COX2, COX3 on the X-axis instead of CI, CII, and CIII to avoid confusion with Complex I, Complex II, and Complex III.

Thank you for the suggestions. We have changed the labelling.

(5) Figure 1d: The font for the individual tRNAs are hard to read. Could you just write the amino acid (e.g. Ala) and above write “tRNA”?

Thank you for this suggestion, we have changed the figure accordingly.

(6) Line 136: “accumulated to a large proportion of the total reads” – can you indicate the proportion here in parentheses?

We have reanalysed the nanopore RNA data and present a more detailed overview of the proportions of different polycistrons in our different models and genotypes (see Figure 2, S3, and S4). We have also added additional text to that result section (line 138-190).

(7) Figure 2a: The yellow chosen here for close to 0 is very hard to see, in particular when printed. Also, I'm not sure I understand what the legend means by "of first transcript".

We have changed the background colour to better visualise the result and defined the "of first transcript" as "Shown are the fractions of transcripts passing through at least the centre of one tRNA, calculated relative to reads in the next mRNA or rRNA." We have clarified this in the figure legend for now figure 3b (line 1195f).

(8) Figure 2b-d: The "Fraction of reads" line thickness bar doesn't provide much information here since the lines here are roughly the same size. I believe this is used here to match Supplementary Figures 3, 4, and 5. That plotting works better in the Supplement where the figures are larger. For the main figure, is it possible to just do fraction of reads? You could also limit the color scale here to not go all the way from 0 – 1 if you're trying to make a point about the fraction of reads that match each of the illustrated processed forms.

Thank you for the suggestion. We agree with the reviewer and have revised the figure to better represent the data, as suggested.

(9) Figure 3d: It's hard to visually process all of the changes here because of the number of samples and proteins being blotted. Can you either (a) separate 8wk, 12wk, and MEF or (b) quantify the bands and plot as a bar graph?

We have added shaded background to separate the groups from each other and hope that this makes it easier to appreciate the differences between the groups. The new figure is now Figure 4d.

Reviewer #2 (Remarks to the Author):

The manuscript by Glasgow et al. provides a very elegant work on the role of methylation for mitochondrial gene expression. The authors established a conditional skeletal muscle-specific mouse model and mouse embryonic fibroblasts (MEFs) deficient in mitochondrial methylation by deleting the mitochondrial SAM carrier (SAMC). These tools allow the authors to investigate the vital role of methylation *in vivo* and *in vitro* specifically within mitochondria without affecting the responsible methyltransferases. The authors convincingly show that mitochondrial RNA precursors are not efficiently processed without SAM especially the gene cluster containing tRNA^{Val}-12S-tRNA^{Phe}-16S leading to an accumulation of rRNA precursors. Interestingly, those rRNA precursors seem to be associated with processing and ribosome biogenesis factors as indicated by sucrose gradient analysis. Using a combination of biochemical approaches, mass spectrometry and cryo-electron microscopy Glasgow et al. reveal the consequences of mitochondrial SAM depletion on mitoribosome assembly. Ten immature mtLSU complexes were solved with different assembly factors bound. This set of biogenesis intermediates reveal alternative assembly pathways. Interestingly, murine mtLSUs contain equally mt-tRNA^{Val} and mt-tRNA^{Phe}, however, the immature mtLSU complexes accumulating in the knockout reveal a shift towards mt-tRNA^{Phe} lacking CCA addition at the 3' end indicating that this modification is not required for the incorporation of the structural tRNA at the CP.

The manuscript is very well written and the figures are of high quality. The study will be of high interest for a broad readership and is suitable for publication in Nature Communications. I have only a few points, which the authors may want to address:

We thank the reviewer for their kind words and helping us to improve our manuscript.

Major comments:

1. Supplementary Fig. 9a / page 6, line 172-174

The authors claim that the sedimentation pattern is normal and that the modified tRNA composition is the reason for the translation defect in Skm-KO. However, in the opinion of the reviewer there are differences between Ctrl and KO. The Ctrl sample shows defined peaks for SSU, LSU and monosome, whereas the signals for uS15m and mL45 tend to tail through the gradient from the SSU and LSU fractions without the defined peak in the monosome fraction. This indicates a problem in monosome formation often accommodated with elevated signals of MRPs. This is not in conflict with the story, it rather supports the vital role of methylation for ribosome biogenesis *in vivo*.

Thank you for pointing this out to us. We have revised the text to reflect the observed shifts in sedimentation patterns (line 228-233).

2. Fig. 3b / page 6, line 174-175

The authors claim "normal signals corresponding to the mtLSU and mtSSU" in MEFs KO. However, there is a clear shift of the signals for mL37 and uS15m by one fraction, which already indicates a defect in subunit biogenesis, which is in agreement with the solved immature LSU complexes. Also here, the authors are encouraged to put some more weight on their biochemical data.

We agree with the reviewer and have changed the text to reflect the observed shifts (233-236).

3. Suppl. Fig. 9g

Change “Samc MEF Ctrl” to “Ctrl MEF” and “Samc MEF KO” to “KO MEF” to be consistent throughout the manuscript

The results presented in Suppl. Fig. 9g looks different to the results in Fig. 3b. Independent of the different gradient conditions, in Suppl. Fig. 9g monosomes seem to be present in contrast to Fig. 3b. The authors may want to revisit the figure.

Thank you for pointing this out. We have changed the labelling.

We agree with the reviewer that some monosome can be seen in KO MEF samples in Figure S9. In agreement with the comment above, figures 4b (fraction 15 versus 17) and S10c (fractions 14-15 versus 15-16) depict a shift in monosome to a lighter fraction, suggesting that both experiments do show similar results. Considering that different numbers of fractions were taken for the two experiments and different antibodies used against LSU (L37 versus L45), we argue that the two experiments still present the same defect in monosome formation. Additionally, the gradients in Figure S9 were slightly overloaded to ensure signal from RPU4 and ERAL1, and the SILAC experiments suggest an accumulation of partially assembled, unprocessed intermediates in the heavier fractions. We clarified this in the legend of Figure S10c.

minor points

4. page 7, line 213

“...LSU and the monosome (fractions 12-15) (Fig. 4d).”

I think this should be changed as followed: “...LSU (fractions 10-12) and the monosome (fractions 13-15)”

Done.

5. page 8, line 232

change “16s” to “16S”

Done.

6. page 9, line 259

delete “ref”

Done.

7. page 10, line 287-288

remove links and delete “ref”

Done.

Reviewer #3 (Remarks to the Author):

The mitochondrial methylation potential gates mitoribosome assembly by Glasgow et al.

The authors investigate the role of S-adenosyl methionine (SAM) in mitochondrial gene expression, focusing on its impact on mitochondrial ribosomal RNA (rRNA) processing and ribosome assembly. Using mouse embryonic fibroblast (MEF) cells depleted of the SAM carrier (SAMC) and quadriceps cells from mice with a skeletal muscle-specific deletion of *Slc25a26* (encoding SAMC), they employ long-read RNA sequencing, quantitative mass spectrometry (label-free and stable isotope labeling), sucrose gradient sedimentation, and cryo-EM to assess the consequences of SAM depletion.

Their findings demonstrate that mitoSAM deficiency disrupts mitochondrial ribosome assembly by impairing rRNA processing, stalling subunit maturation, affecting structural tRNA incorporation, and preventing proper peptidyl transferase center (PTC) formation. These results underscore the critical role of mitoSAM in mitochondrial gene expression and ribosome biogenesis.

While the study is methodologically sound and presents compelling data, the authors could better incorporate and discuss relevant previous research in this area.

We thank the reviewer for their time, helpful links to previous work, and the detailed proofreading of our manuscript. It was much appreciated.

Major Points:

There are several relevant references that I would like to bring to the authors' attention. For example, Ishiguro (2019, NAR) showed that SAM depletion in *E. coli* causes a growth defect that can be rescued by overexpression of the methyltransferase RlmE, the likely bacterial homolog of MRM2. Similarly, Arai et al. (DOI: 10.1073/pnas.1506749112) and Wang et al. (10.1073/pnas.1914323117) described structural and mechanistic consequences of RlmE depletion, resulting in LSU precursors with structural deficits comparable to those reported here. Additionally, Nikolay et al. (10.1016/j.molcel.2021.02.006) identified a natural LSU precursor containing ObgE - GTPBP10 is one of the two ObgE homologs - with incomplete maturation in similar regions of 23S rRNA (H90-93, H89, H80). Finally, Lai et al. (10.1371/journal.pone.0090818) identified *Ftsj2* (MRM2, Uniprot ID Q9CPY0) as the mitochondrial homolog of RlmE.

The authors may incorporate these references along with any other relevant studies and discuss how their findings align with or extend prior work. The discussion is currently brief and would benefit from a more thorough comparison with these studies.

Thank you for pointing out these references. We have integrated these references into the discussion, highlighting their relevance to our findings

Minor Points:

The phrasing in some instances should be clearer and more precise:

- Line 31: "...required during peptidyl transferase center formation and mitochondrial ribosome assembly." → PTC formation is part of ribosome assembly. This should be reworded for accuracy.

We have reworded to clarify that PTC formation is a step within ribosome assembly (Line 37ff).

- Line 32: "Our data thus identify a critical role for methylation at two steps during mitochondrial gene expression." → What are these two steps? This should be explicitly stated.

We have reworded the abstract to clarify this.

- Line 74: "... and the deletion of SAMC in mice and flies is embryonic and larvae lethal. → The sentence is slightly awkward. Please rephrase e.g., "...and SAMC depletion in mice and flies results in embryonic and larval lethality.

We have changed the wording (Line 87f).

- Line 93: "Born at Mendelian ratios and appeared normal at weaning but needed to be sacrificed by 12 weeks of age due to plateaued weight." → This should be reworded for clarity and provide more context for readers unfamiliar with mouse development.

We clarified that KO mice exhibited halted weight gain weight, compared to their littermates, necessitating ethical sacrifice at ~15% of weight loss (107-109).

- Line 96: "Quadriceps presented with less than 50% methylation." → Consider rephrasing as: "The quadriceps muscle exhibited a 12S rRNA methylation level of less than 50%."

We have reworded the text (Line 111f).

- Line 100: Please define OXPHOS.

We have now defined OXPHOS (Line 116).

- Lines 121ff: What causes the accumulation of incomplete transcripts? Why do they pile up?

Our manuscript demonstrates that depletion of mitochondrial SAM leads to the accumulation of unprocessed transcripts, with a failure to efficiently cleave a subset of tRNA-containing junctions. Based on our data, we propose that these specific junctions require mitoSAM for proper processing. The methyltransferase TRMT10C (MRPP1), together with SDR5C1 (MRPP2), is thought to form a structural platform that facilitates cleavage by either PRORP (MRPP3) or ELAC2 at the 5' or 3' side of the tRNA. Our findings suggest that, contrary to previous reports, MRPP1 activity is indeed necessary for the processing of certain mitochondrial tRNA junctions. We have expanded on this in the discussion.

- Line 160: Please specify which mitochondrial transcription inhibitor was used.

We used the mitochondrial transcription inhibitor version 1B (IMT1B). Bonekamp, N. A. et al. Small-molecule inhibitors of human mitochondrial DNA transcription. *Nature* 588, 712–716 (2020). We have clarified this throughout the manuscript and added the appropriate reference.

- Line 164: "Thus, this data identify methylation within the rRNA gene cluster as critical for efficient processing." → Does this mean "critical for its efficient processing"? Please clarify.

We revised the text to clarify that methylation is critical for efficient rRNA processing (Line 222).

- Line 171: "We performed sedimentation experiments on the mitochondrial ribosome." → This phrasing is awkward; please reword.

We have rewritten the relevant section (228-236).

- Line 205: "Additionally, early processing and maturation factors in the monosome fractions indicate that a larger protein-RNA complex already assembles prior to complete cleavage of the ribosomal gene cluster." → This sentence is unclear; please rephrase.

We have clarified the statement (Line 268-272).

- Line 221: "Thus, our results indicate that in the absence of mitoSAM, 12S and 16S rRNAs initiate early stages of assembly but stall due to a lack of processing." → A lack of processing of what? Please specify.

We revised the text to specify the lack of rRNA processing (Line 287f).

- Line 273: "GTPBP7 was bound in a fluid conformation..." → "Fluid conformation" is vague; consider using more precise terminology.

We have re-written the section. (lines 297-323)

- Line 291: "Methylation is of particular importance at two sites during mitochondrial gene expression, with both rRNAs and tRNAs possessing several modifications." → What are these two sites? This should be specified.

We have re-written the discussion (Line 385f).

- Line 310: "Our findings also show that the 2'-O-methylations of 16S rRNA are crucial for the maturation of the PTC and LSU. → Similar to Line 31. Please rephrase, as the PTC is part of the LSU.

We have re-written the discussion (Line 391-393).

- Figure Labels: In several figures (e.g., Fig. 2a, e; Fig. 4d; Fig. 5; Fig. 6b; Supp. Fig. 9e, f), the labeling text is too small. Ensure that the final scaled version maintains a minimum font size of 6 points.

We have adjusted the text size and hope that in the final version the font will be big enough.

- Fig. 6b: Verify and correct the color coding for the A loop and P loop.

Thank you for noticing. We have corrected the colours.

- Fig. 9e, f: These Figs. are overcrowded and difficult to interpret. Consider an alternative visualization or highlighting only a subset of peptides/ proteins. Also, were these analyses repeated? If so, indicate sample size (n=x).

We agree with the reviewer and now split the figure into two. This should allow for a clearer visualisation. We have also indicated the sample size.

- Supplementary Figures: Many supplementary figures, particularly Supp. Figs. 3–8, have insufficient legends. More detailed figure legends should be provided.

We have expanded on the figure legends for the supplementary figures.

Reviewer #4 (Remarks to the Author):

Summary:

The manuscript by Glasgow et al. proposes two roles for SAM in mitochondrial gene regulation. Direct long-read RNA sequencing highlights the requirement for SAM in efficient RNA processing (particularly the rRNA cluster). Profiling of ribosome fractions identifies the enrichment of RNA processing factors and mitoribosome assembly factors suggesting a defect in mitoribosome production. Cryo-EM studies reveal 10 distinct intermediate assemblies (immature LSU classes) where notably the peptidyl transferase center appears disordered. This manuscript was very well written. While the results presented in this work are important for the field, there is key information and context missing. This is particularly true when considering the relative low resolution of the cryoEM data and the associated assertions made on the molecular basis of LSU maturation.

We thank the reviewer for their time and advise.

Please find specific comments below.

1.The authors place a strong focus on tRNA containing processing sites (likely due to the known methylation of mito-tRNAs). However, there is a short statement in the discussion that there was no impact on atypical processing sites (sites without tRNAs). This selectivity appears to be an important point and the data should be incorporated into a Supplementary Figure.

We have reanalysed the nanopore data and show 5'and 3'processing sites across the mitochondrial transcriptome (New figures 2 and S4), which clearly show abnormal processing at the specific canonical sites and normal at the non-canonical sites. We have added this result to the manuscript (Line 146-190).

2.The authors discuss the role of SAM in processing efficiency, but do not comment on observed changes in processing fidelity or alternative processing sites. The authors should comment on whether SAM exclusively impacts cleavage efficiency of canonical sites or whether other spurious RNA processing events may also influence translation dysregulation.

Thank you for this suggestion. We have re-analysed the ONT sequencing data and observe no new spurious processing sites (See Figure S3).

3.The authors state in the introduction that methylation of mito-tRNAs is not necessary for precursor RNA processing by RNase P and RNase Z. Therefore, the molecular mechanism remains unclear. Did the authors measure RNase P/Z abundance to see whether changes in SAM import impact the levels of these endoribonucleases?

Our proteomic analysis identified several components of mtRNase P and mtRNase Z (ELAC2), with no difference in abundance between control and KO samples, suggesting that the reduced processing efficiency is not caused by a lack of appropriate endonucleases.

4.The manuscript lacks important information about the quality of the structural data presented in this work. For instance, Supplementary Figures 10 and 11 demonstrate a clear distinction in the map quality and interpretability across the 10 reconstructions. The authors need to include a description of the overall quality of these maps in the results section prior to discussing their interpretations. In areas of particular focus (e.g. PTC, CP), the authors should

describe the local resolution and what can and cannot be inferred from these regions of the map.

We added a detailed description of map quality, including resolution ranges and local resolution for key regions (e.g., CP, PTC), in the Results section. Specifically, we describe the resolution ranges of different states and highlight the local resolution in key regions, such as the CP and PTC. CP-tRNA and interface rRNA/PTC local resolution figures have been added to Supplementary Fig. 12. Details regarding resolution and model building for specific states have also been updated in the methods. This addition clarifies the level of detail that can be inferred from these regions of the maps. The revised text can be found in the Results section (lines 297-323).

5. Supplementary Table 12 is missing from the submitted files. This is imperative for readers to discern the confidence one can place on the conclusions made from each map/model.

We apologise for the confusion. Table 12 is now among the uploaded files.

6. The authors state “Surprisingly, no 3'CCA RNA modification was present in all mature tRNAs, suggesting that this modification is not required for incorporation into the mitochondrial ribosome” (lines 239-241). This conclusion appears to be motivated by the lack of density at the 3'-end of the tRNA presented in Figure 5b. The authors need to specify which map is displayed and what is the local resolution in this area of the map. The resolution appears insufficient to make such a definitive claim.

We provide details in Supplementary Figures 11 and 12, including local resolution (2.9–5.5 Å) and acknowledge limitations in interpreting tRNA modifications. (line 333-341).

7. Similar to point 6, the authors state in lines 231-233 that the cryo-EM maps lack features to describe methylation groups. While this may be true, the resolution in this region of the map appears to be insufficient to state definitively that there are no RNA modifications. The authors should be cautious in this interpretation.

We have changed the relevant section in the text and added the local resolution to emphasise our observation (line 336).

8. The authors leverage chemical crosslinking to stabilize LSU intermediates. The authors should include this important point in the results and demonstrate how they optimized or verified the crosslinker is not influencing observed distortions.

We now include more information in the results section (line 305-312) and material and methods section (line 801-805).

9. Were the authors able to observe SSU intermediates? It would be informative to know whether SAM also impacts SSU maturation and how.

We fully agree with the reviewer and initially intended to report structures for SSU and LSU. However, sample and grid preparation heavily enriched for LSU samples, with only a small number of SSU subunits, which were insufficient to obtain any reasonable structures from. We, however, intend to modify our protocols to enrich also for SSU in the future.

Minor points:

Please label Figure 2, panel E.

We have labelled the figure.

Supplementary table listing SamC mouse genotyping primers appears incomplete. Please revise.

We have added the primer sequences. Thank you for noticing.

Supplementary Figure 10 has replicated typos “re finement”

Done.

RESPONSE TO REVIEWERS' COMMENTS

We once again would like to thank the reviewers for their support and helpful comments to improve this manuscript.

Reviewer #1 (Remarks to the Author):

The authors have addressed my concerns. Their re-analysis of their ONT data clearly shows the effects on the individual processing sites and could be a useful pipeline for further studies dissecting the roles of specific modifications on mtRNA processing. They also expanded analysis and discussion of mitoribosome assembly defects and compared their results to phenomena observed in *E. coli*. I think the manuscript will be of great interest to the fields. I have only a few minor comments:

- The title and y-axis label in Figure 2C could be more streamlined. Potentially “Unprocessed read differences (KO-Ctrl)” for the title and “% difference” or “Percent difference” for the Y-axis.

We have implemented these suggestions.

- Supplementary Figure 5b: It would be nice to have a table title so it's understandable without the caption

We have added a title.

- Line 298: “CP” introduced but not defined as “central protuberance”

We have added the definition.

We thank the reviewer for their time and help.

Reviewer #2 (Remarks to the Author):

The authors have addressed all issues. In my opinion the revised manuscript is suitable for publication in Nature Communications.

We thank the reviewer for their time and help.

Reviewer #3 (Remarks to the Author):

The authors have done a great job improving their manuscript, including addressing the comments and suggestions from the other reviewers, which— from my perspective— were handled very well.

While reviewing the revised version, I noticed a minor typo:

- Fig. 6b: acceter arm → acceptor arm

We have corrected the typo.

Overall, the scientific quality, data presentation, and readability have improved significantly. In my opinion, this is a very nice piece of work that deserves publication in Nature Communications.

We thank the reviewer for their time and help.

Reviewer #4 (Remarks to the Author):

The authors have revised the manuscript by Glasgow et al. to address the concerns and suggestions raised by the reviewer. The major changes include a reanalysis of ONT sequencing data to provide additional insight into methylation-dependent RNA processing

and an improved description of the quality and interpretability of the cryoEM reconstructions. The revised text and new figures present a compelling story that is important for the field. The authors addressed concerns related to the structural study, and I support its publication.

We thank the reviewer for their time and help.